# Which Heads Matter for Reasoning? RL-Guided KV Cache Compression

Wenjie Du[1]  Li Jiang[2 3]  Keda Tao[1 4]  Xue Liu[2 5]  Huan Wang[1]

https://kurt232.github.io/RLKV

## Abstract

Reasoning large language models exhibit complex reasoning behaviors via extended chain-of-thought generation that are highly fragile to information loss during decoding, creating critical challenges for KV cache compression. Existing token-dropping methods directly disrupt reasoning chains by removing intermediate steps, while head-reallocation methods, designed for retrieval tasks, fail to preserve the heads essential for generative reasoning. However, no existing method can identify which attention heads genuinely maintain reasoning consistency and control generation termination. To address this, we propose RLKV, which uses reinforcement learning as a probe to discover which heads contribute to reasoning quality by directly optimizing their cache usage against actual generation outcomes. This discovery naturally leads to an efficient compression strategy: we allocate full KV cache to reasoning-critical heads while aggressively compressing others with constant-size KV cache. Experiments reveal that a fraction of heads proves essential for reasoning, enabling **20–60%** cache reduction with near-lossless performance across diverse tasks and models, and up to **2.06×** end-to-end speedup at 60% reduction.

## 1. Introduction

Recent advanced reasoning large language models (LLMs) (Jaech et al., 2024; Team et al., 2025; Guo et al., 2025; Google DeepMind, 2025) exhibit complex reasoning behaviors, such as self-reflection to revisit previous steps and exploration of alternative approaches, and achieve revolutionary performance on challenging mathematical and cod-

[1]Westlake University [2]McGill University [3]Mila - Quebec AI Institute [4]Zhejiang University [5]Mohamed bin Zayed University of Artificial Intelligence. Correspondence to: Huan Wang <wanghuan@westlake.edu.cn>.

*Proceedings of the 43$^{rd}$ International Conference on Machine Learning*, Seoul, South Korea. PMLR 306, 2026. Copyright 2026 by the author(s).

ing problems. However, this breakthrough comes at a cost: the extended chain-of-thought (CoT) reasoning generates significantly more tokens compared to instruct models, creating substantial deployment challenges. More critically, these extended reasoning chains prove highly fragile to information loss, where KV cache compression methods that work well for instruct models severely degrade reasoning scenarios.

As illustrated in Figure 1 (a), existing KV cache compression methods typically follow one of two strategies: token dropping or head reallocation. Token-dropping methods selectively evict less important tokens from each head's KV cache (Zhang et al., 2023; Li et al., 2024; Cai et al., 2025; Yang et al., 2024b; Qin et al., 2025), while head-reallocation methods identify critical heads and allocate full KV cache to them, applying compressed KV cache to the remaining heads. However, as shown in Figure 1 (b, left), two representative methods, including token-dropping method R-KV (Cai et al., 2025) and head-reallocation method DuoAttention (Xiao et al., 2025), degrade significantly when applied to reasoning models, while maintaining stable performance on their instruct counterparts. In the MBPP (Austin et al., 2021) coding task, both model variants achieve nearly identical uncompressed performance. This controlled comparison isolates extended CoT generation as the primary cause of compression challenges, rather than differences in model capability. In reasoning models, the KV cache serves not merely as computational optimization but as the carrier of reasoning behaviors, storing critical states for CoT consistency and controlling generation flow. This fundamental shift raises a critical question: **which KV attention heads matter for reasoning behaviors?**

Existing methods fail to answer this question due to fundamental limitations in their design. As illustrated in Figure 1 (b, right), the two compression strategies exhibit distinct error modes as compression rates increase. Models with token-dropping methods (R-KV) apply compression uniformly across all heads by selectively evicting tokens, inevitably discarding reasoning-critical information that disrupts CoT consistency and leads to repetitive loops that fail to progress toward solutions. Although the R-KV approach (Cai et al., 2025) is designed specifically for reasoning models, it still cannot escape this inherent limitation. In contrast,

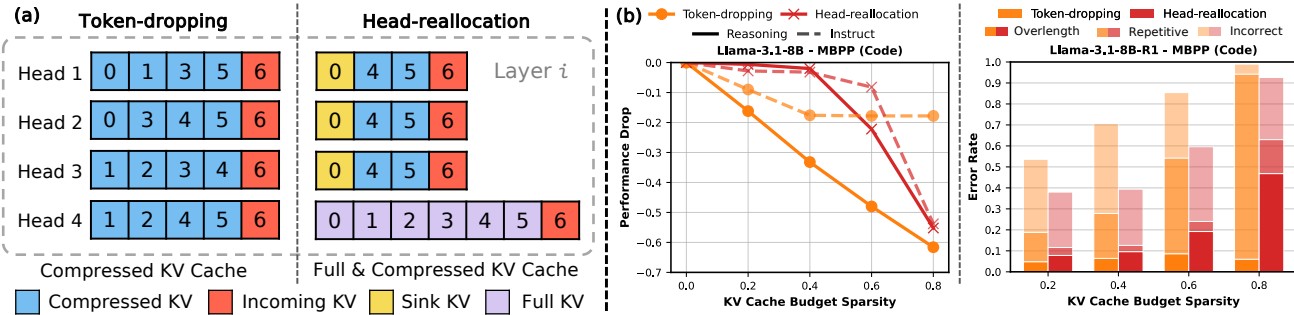

*Figure 1.* **(a) Overviews of Two Methods** *Left:* Token-dropping method removes less important tokens from each head's KV cache. *Right:* Head-reallocation method allocates full KV cache to critical heads while assigning constant-size KV cache to the remaining heads. **(b) Case study.** *Left:* The token-dropping method (R-KV) and the head-reallocation method (DuoAttention) maintain relatively stable performance on Llama-3.1-8B-Inst but degrade substantially on Llama-3.1-8B-R1, largely due to the longer generations produced by the reasoning model. *Right:* In terms of error modes, the token-dropping method (R-KV) tends to degenerate into repetitive behavior whereas the head-reallocation method (DuoAttention) often produces over-extended CoT that exhausts the length budget without reaching a correct solution. See Appendix A for complete results.

models with head-reallocation compression preserve complete sequence information in selected heads by allocating full KV cache to them while compressing others. This approach maintains more coherent reasoning than token-dropping, but remains ineffective: for problems that the uncompressed model can solve, the compressed model goes astray in its reasoning process and is unable to reach a solution within the maximum budget. This failure stems from their head identification mechanisms, which target retrieval heads (Wu et al., 2025) for recall tasks. This motivates our key insight: **identifying reasoning-critical heads requires directly observing how each head's compression affects actual reasoning outcomes during generation.**

To achieve this, we propose RLKV, which employs reinforcement learning (RL) as a probe to directly observe the relationship between each head's KV cache compression and reasoning quality. As illustrated in Figure 2, our method generates reasoning samples during RL training and assigns rewards based on their quality. These reward signals guide the optimization of learnable gating adapters that control the mixing of full attention and compressed local attention for each head, with L1 penalty encouraging sparsity. Through this RL optimization process, the learned gating scores reveal a critical insight: only a small subset of heads requires full KV cache to maintain reasoning consistency, while others can be aggressively compressed without performance loss. We term these heads requiring full cache as **reasoning-critical heads**. Our method naturally translates this finding into an efficient compression strategy: we allocate full KV cache to reasoning-critical heads while applying compressed constant KV cache to others, effectively preserving reasoning behaviors during inference.

Our work makes three main contributions: **First**, we introduce RLKV, a novel method that employs lightweight reinforcement learning as a probe to identify reasoning-critical heads. It functions by directly observing how cache

compression impacts reasoning quality during generation. **Second**, RLKV achieves state-of-the-art compression performance, enabling near-lossless reasoning with a 20–60% reduction in KV cache usage across diverse tasks and models, and delivers up to a **2.06×** end-to-end speedup at 60% reduction under SGLang continuous batching. **Third**, to our knowledge, RLKV is the first to identify specific attention heads essential for reasoning. Through comprehensive analyses of performance, head sensitivity, error modes, and response length, we provide a new perspective on understanding reasoning models from a KV cache compression viewpoint.

## 2. Related Work

**Efficient LLM Inference.** Various techniques reduce KV cache overhead through architectural or system optimizations. Grouped-Query Attention (GQA) (Ainslie et al., 2023) and Multi-head Latent Attention (MLA) (Liu et al., 2024a) reduce the number of KV heads by sharing them across query heads, achieving significant memory reduction but requiring expensive pre-training from scratch. Linear attention methods (Gu & Dao, 2024; Yang et al., 2025b) maintain constant memory usage during inference by avoiding the quadratic attention computation, but exhibit reduced modeling capacity compared to standard transformer architectures. KV cache quantization (Liu et al., 2024b; Tao et al., 2025b; Hooper et al., 2024; Duanmu et al., 2024; Su et al., 2025; Yue et al., 2024) and system-level optimizations, such as paged KV cache (Kwon et al., 2023), KV cache reuse (Zheng et al., 2024), and sparsely loading KV cache (Tang et al., 2024), provide orthogonal improvements by reducing the precision or optimizing the storage and retrieval methods of cached states. While sparse attention methods (Child et al., 2019; Beltagy et al., 2020; Lu et al., 2025; Yuan et al., 2025) further accelerate inference by utilizing intra-head sparsity, they often still require full KV cache storage.

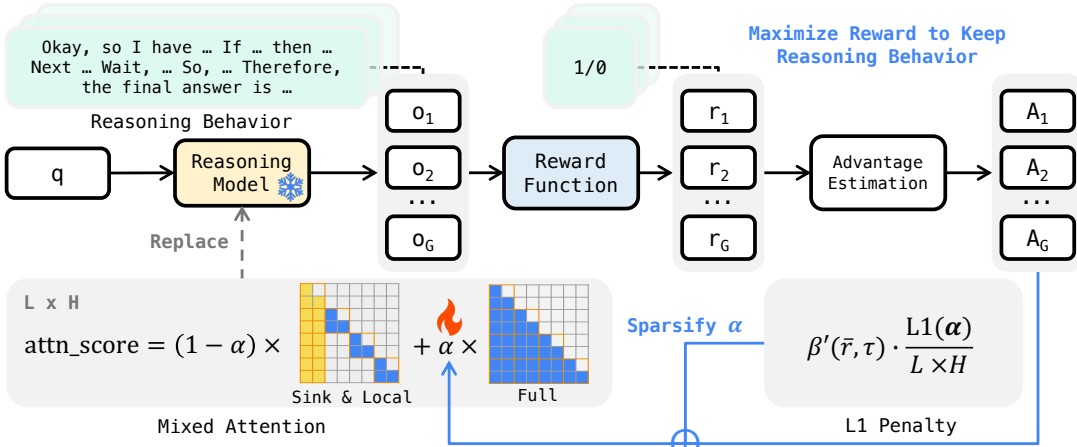

*Figure 2.* **Overview of RLKV:** Our method proposes to utilize RL as a probe to identify reasoning-critical heads. The RL pipeline naturally captures reasoning behaviors, since it samples the current model's generations to produce reward signals. The reward function evaluates the samples to assess reasoning quality. We employ $L \times H$ learnable gating adapters to mix full attention and local attention for each head, quantifying each head's reliance on full versus local KV cache access. We apply an L1 penalty to encourage adapter sparsity, while RL optimizes the adapters to preserve reasoning behaviors. After training, we identify reasoning-critical heads with high adapter values and allocate full KV cache to them while applying compressed KV cache to others for efficient inference.

Ultimately, these methods treat KV cache as opaque data without exploiting the inherent head-level sparsity patterns.

**KV Cache Compression.** Recent works mainly exploit sparsity in long-context scenarios for instruct models, including token-dropping and head-reallocation methods. (1) Token-dropping methods (Zhang et al., 2023; Li et al., 2024; Cai et al., 2025; Yang et al., 2024b; Qin et al., 2025) apply eviction strategies across all heads or intra-layer heads based on attention scores. $H_2O$ (Zhang et al., 2023) maintains important tokens' KV cache based on accumulated attention scores plus a sliding window for recent tokens. Specifically, recent R-KV (Cai et al., 2025), designed for reasoning models, primarily adds similarity-based clustering to priority evict redundancy tokens' KV cache during both prefill and decoding phases. However, they inevitably discard reasoning-critical information and disrupt the CoT consistency as compression rates increase. (2) head-reallocation methods maintain full KV cache only for identified retrieval heads (Wu et al., 2025) in long-context scenarios while applying compressed KV cache to others. Some methods(Fu et al., 2024; Tang et al., 2025; Kim et al., 2025) use proxy metrics of attention scores for head identification, while DuoAttention (Xiao et al., 2025) and PruLong (Bhaskar et al., 2025) are learning-based methods. DuoAttention minimizes single-forward output deviation on a synthetic long-context recall task, while PruLong uses next-token loss on long-context pre-training corpora. However, these methods cannot capture the reasoning behaviors that emerge during dynamically extending CoT generation, as their static heuristics or teacher-forced objectives miss how compression errors accumulate along the autoregressive trajectory.

**Reinforcement Learning for Efficiency.** RL has proven effective in Neural architecture search (Zoph & Le, 2017;

Zoph et al., 2018), where it treats architecture choices as sequential decisions, and model pruning (He et al., 2018), where it learns layer-wise pruning ratios that maximize accuracy under resource constraints. However, the limitation is the high computational cost due to the large optimization space. Our work utilizes gating values assigned to each KV head to reduce the optimization space and make RL feasible and efficient. For reasoning language models, recent works apply RL tuning to mitigate overthinking (Hou et al., 2025; Liu et al., 2025b) by learning to reduce CoT length while maintaining reasoning capability, thereby indirectly decreasing KV cache requirements. Our work is orthogonal to these methods, employing lightweight RL training to identify reasoning-critical heads that guide KV cache compression while preserving reasoning capability.

## 3. Methodology

In this section, we present RLKV, a novel reasoning-critical head identification method to guide efficient KV cache compression for reasoning LLMs, as illustrated in Figure 2. We operationally define "**reasoning-critical heads**" as KV heads that significantly degrade reasoning under local KV cache access; these require a full KV cache to maintain CoT consistency, while others are compressible. To achieve this, we follow prior head-reallocation methods to use mixed attention with gating adapters to quantify each head's reliance on complete or compressed KV cache usage. Then we apply RL with sparsity pressure to optimize the gating adapters based on a verifiable reward signal, naturally capturing reasoning behaviors. Finally, we introduce two complementary stabilization techniques to address the conflict between dense regularization and sparse rewards as the sparsity increases.

## 3.1. Mixed Attention with Gating Adapters

Identifying reasoning-critical heads requires estimating the sensitivity to complete KV cache usage of individual KV heads; therefore, we employ an extra gating parameter $\alpha_{l,h} \in [0,1]^{L \times H}$ to each head $h$ and each layer $l$ after scaled dot product attention(Xiao et al., 2025; Tang et al., 2025; Bhaskar et al., 2025). And we can construct the full KV cache and local KV cache access via attention mask ($\mathbf{M}_{casual}$ and $\mathbf{M}_{local}$):

$$\begin{aligned} \text{out\_mix\_attn}_{i,j} = \alpha_{i,j} \cdot \text{out\_full\_attn} + \\ (1 - \alpha_{i,j}) \cdot \text{out\_local\_attn} \end{aligned} \quad (1)$$

which uses lightweight gating adapters to quantify each head's reliance on full versus local KV cache access. We use the local attention mask (Xiao et al., 2024) with the constant initial sink tokens and recent tokens for numeric stability. This design dramatically reduces the optimization space to only $L \times H$ gating parameters by freezing all LLM parameters, making it feasible to apply RL for identifying reasoning-critical heads.

## 3.2. RL for Reasoning Head Identification



*Figure 3.* Gating score distribution after RLKV training on three models, all of which adopt the GQA architecture. Qwen-2.5-7B-R1 exhibits inherent limitations in achievable sparsification without compromising reasoning behavior, due to its larger KV group size of 7 (compared to 4 in other models).

Reasoning LLMs are often post-trained using reinforcement learning with verifiable reward (RLVR) (Guo et al., 2025; Team et al., 2025), which enhances reasoning capabilities by evaluating generated samples based solely on final answer correctness. During this RL training process, reasoning behaviors are naturally exhibited in the sampled CoT sequences, while reward signals directly reflect reasoning quality. These two characteristics make RLVR ideal for reasoning-critical heads identification.

In concrete, we optimize the gating adapters $\boldsymbol{\alpha}$ using Group Relative Policy Optimization (GRPO) (Shao et al., 2024)

on mathematical reasoning problems with two key modifications. First, to maximize the discriminative power of reward signals for *reasoning head* identification, we remove the KL penalty that conventionally limits reward signal strength to prevent over-optimization, as RLKV freezes all LLM parameters and only optimizes the gating adapters. Second, we apply L1 regularization (Tibshirani, 1996) to the adapters by incorporating the scaled L1 penalty term $\beta = \|\boldsymbol{\alpha}\|_1/(L \times H)$into the objective function to encourage adapter sparsity. The reward signal preserves high $\alpha_{i,j}$ values for reasoning-critical heads requiring full KV cache access, while the L1 penalty drives $\alpha_{i,j}$ toward 0 for compressible heads.

The overall objective is defined to maximize:

$$\underbrace{\frac{1}{G}\sum_{i=1}^{G}\min\left(\frac{\pi_{\boldsymbol{\alpha}}(o_i|q)}{\pi_{\boldsymbol{\alpha}_{\text{old}}}(o_i|q)}A_i, \text{clip}\left(\frac{\pi_{\boldsymbol{\alpha}}(o_i|q)}{\pi_{\boldsymbol{\alpha}_{\text{old}}}(o_i|q)}, 1-\epsilon, 1+\epsilon\right)A_i\right)}_{\text{reward signal}} - \underbrace{\frac{\beta}{L \times H}\|\boldsymbol{\alpha}\|_1}_{\text{L1 penalty}}$$
$$(2)$$

where $q$ is the input query, $\{o_i\}_{i=1}^{G}$ are sampled outputs, $A_i$ is the normalized advantage, computed using a group of rewards $\{r_1, r_2, \cdots, r_G\}$ tailored to outputs:

$$A_i = \frac{r_i - \text{mean}(r_1, r_2, \cdots, r_G)}{\text{std}(r_1, r_2, \cdots, r_G)}. \quad (3)$$

The clipping mechanism with threshold $\epsilon$ prevents excessive policy updates, and $\beta$ controls the regularization strength. The policy $\pi_{\boldsymbol{\alpha}}$ represents the model's generation probability distribution conditioned on the current gating parameters $\boldsymbol{\alpha}$, and the advantage $A_i$ is positive for outputs leading to correct reasoning and negative for incorrect reasoning. This optimization naturally converges to a sparse solution where reasoning-critical heads maintain high $\alpha$ values, as demonstrated in Figure 3.

## 3.3. Stabilization for RL Training

**Sparse Reward versus Dense Penalty**

*Figure 4.* The conflict of sparse reward versus dense penalty leads to training collapse without our stabilization techniques. As adapters become sparse (decreasing average), model performance degrades (dropping reward) and fails to recover.

As adapters become increasingly sparse, the mixed attention

of reasoning-critical heads degenerates to the streaming attention, severely degrading the model's reasoning capacity, as shown in Figure 4. This degradation renders the reward signal increasingly sparse and unstable, while the L1 penalty remains dense across all parameters. This imbalance creates a vicious cycle, where degraded performance leads to sparser rewards, making the dense L1 penalty relatively stronger, which further drives adapters toward zero with no recovery capability. To resolve this destructive training dynamic and stabilize the training process, we introduce two complementary techniques that address this challenge from both the reward and penalty perspectives.

**Self-distillation Sampling.** Overly challenging problems during RL training lead to frequent failures and unstable reward signals. In contrast to typical RLVR that utilizes sparse rewards for capability enhancement, our work leverages RL for capability preservation under sparsity constraints. Consequently, we focus on constructing high-quality training data that produces stable reward signals to improve learning efficiency. We construct training data by first filtering all problems the model initially solves correctly, then curating them to 3k using a curriculum sampling strategy (Team et al., 2025). We use output token lengths as a proxy for difficulty, enabling curriculum control that maintains stable reward signals throughout the training process. See Section 4.1 for training dataset details.

**Adaptive Penalty Weighting.** To address the penalty imbalance, we modulate the scaling weight $\beta$ of the L1 penalty based on the reward signal. Our design incorporates two protective mechanisms to prevent training collapse. First, we use adaptive scaling centered around a target reward of $\bar{r} \approx 0.7$ to smoothly decay penalty when performance degrades and increase it when performance improves. Second, we implement a hard cutoff at threshold $\tau$ to completely eliminate regularization when reasoning capability severely degrades. We implement this through a dynamic weight that replaces the constant hyperparameter $\beta$:

$$
\begin{aligned}
\beta'(\bar{r}, \tau) &= \mathbb{I}(\bar{r} > \tau) \cdot \beta \cdot (\exp(\bar{r}) - 1), \\
\bar{r} &= \text{mean}(r_1, r_2, \cdots, r_G),
\end{aligned}
\tag{4}
$$

where the exponential function $(\exp(\bar{r}) - 1)$ provides the adaptive scaling, and the indicator function $\mathbb{I}(\bar{r} > \tau)$ provides the hard cutoff based on mean reward $\bar{r}$ in the current group.

The end result is a set of identified reasoning-critical heads that require full KV cache access, while less relevant heads can utilize compressed KV cache access, achieving significant memory compression without sacrificing reasoning capability. While training optimizes the continuous gating adapters $\alpha$ via mixed attention, at inference we discretize this gating into a top-$k$ binary selection: heads with the highest $\alpha$ values retain full KV cache access according to

the target compression ratio, while remaining heads still use full attention but with compressed KV cache, which retains only initial sink tokens and recent tokens.

## 4. Experiments

### 4.1. Experimental Settings

**Models, Datasets, and Baselines.** We evaluate RLKV on three mainstream small-scale reasoning models: Llama-3.1-8B-R1 (Guo et al., 2025), Qwen-2.5-7B-R1 (Guo et al., 2025), and Qwen-3-4B-Thinking (Yang et al., 2025a). We conduct experiments on four reasoning benchmarks as well as four subsets of the knowledge QA benchmark MMLU-Pro (Wang et al., 2024). The reasoning benchmarks include three mathematical reasoning datasets—GSM8K (Cobbe et al., 2021) for elementary problems, Math500 (Lightman et al., 2023) for intermediate problems, and AIME24 (Mathematical Association of America, 2024) for advanced problems—to evaluate performance across difficulty levels, together with MBPP (Austin et al., 2021) for Python programming to assess generalization beyond the training domain. To assess generalization beyond the training domain and context length, we additionally evaluate on four MMLU-Pro subsets (Chemistry, Computer Science, Law, Physics; up to 500 randomly sampled instances per subset) and on LongReason (Ling et al., 2025) (64K-input subset, 400 of 794 samples, 70K context). We compare our method with KV cache compression approaches including $H_2O$ (Zhang et al., 2023) and R-KV (Cai et al., 2025), which are typical token-dropping methods, and DuoAttention (Xiao et al., 2025) and KVZip (Kim et al., 2025), which are head-reallocation methods. Given the significant length variation in reasoning tasks (see Appendix E), fixed budgets lead to inconsistent compression ratios. To address this, we adopt $H_2O$ and R-KV with dynamic budgets to ensure fair comparison. As shown in Appendix F, this modification is crucial for fairness and does not penalize the baselines; in fact, it enhances their performance compared to fixed budgets.

**Implementation Details.** We implement RLKV in AReaL(Fu et al., 2025) for RL training and SGLang(Zheng et al., 2024) as the rollout engine, where the attention function is replaced by mixed attention. We optimize gating adapters using GRPO with 4 samples per query and AdamW (Loshchilov & Hutter, 2019) with learning rate 0.01. We filter 3,000 mathematical problems from DeepScaleR (Luo et al., 2025) following our curriculum sampling strategy. And we train the models for 185 steps on 2 NVIDIA A100 GPUs (80GB) for several hours. During training, local attention uses 128 sink tokens and 256 local tokens; during evaluation, we apply the compressed KV cache size with 16 sink tokens and 64 local tokens. We augment all baselines with equivalent token overhead for fair evaluation. Details are provided in Appendix B.

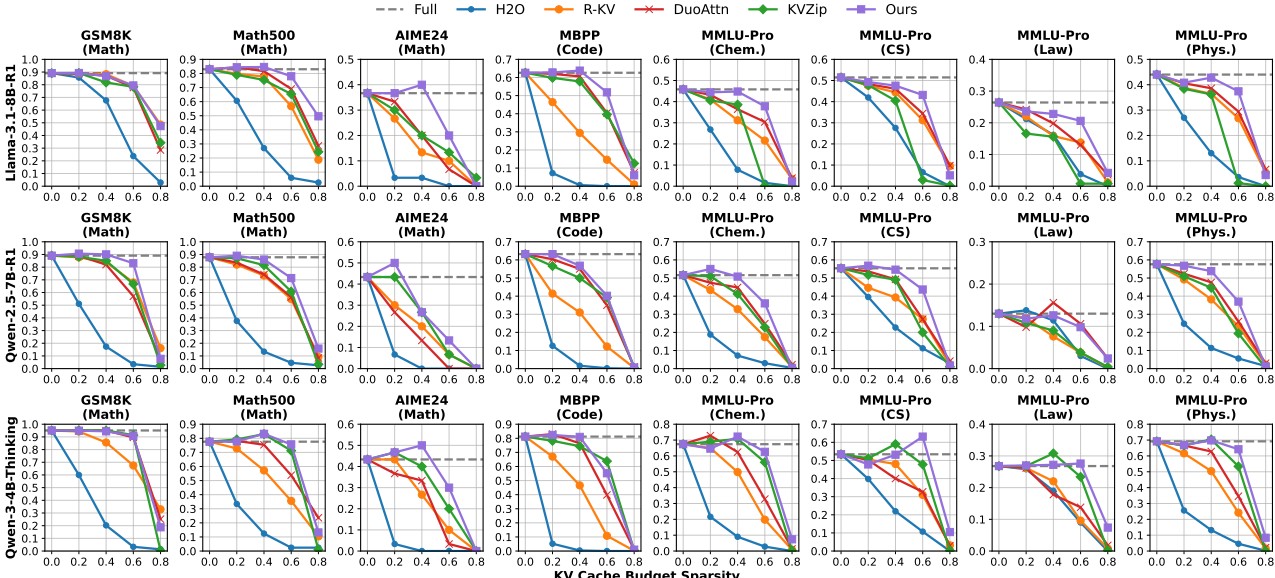

*Figure 5.* **Main Results.** RLKV (Ours) achieves better accuracy-efficiency trade-off across four reasoning benchmarks (GSM8K, MATH, AIME24, MBPP) and four subsets of the knowledge benchmark MMLU-Pro (Chemistry, Computer Science, Law, Physics).

*Table 1.* **Near-Lossless Performance of RLKV.** Each cell shows RLKV's accuracy at near-lossless sparsity threshold, with the delta and that sparsity in parentheses. The threshold varies across benchmarks and models, capturing how task difficulty, type, and the model itself jointly affect RLKV's compression sensitivity.

| Benchmark | Llama-3.1-8B-R1 | Qwen-2.5-7B-R1 | Qwen-3-4B-Thinking |
|---|---|---|---|
| *Math/Code* | | | |
| GSM8K | 86.8 (−2.4, sp=0.4) | 90.1 (+1.0, sp=0.4) | 94.4 (−0.7, sp=0.4) |
| Math500 | 84.6 (+1.6, sp=0.4) | 86.0 (−1.8, sp=0.4) | 75.6 (−2.0, sp=0.6) |
| AIME24 | 40.0 (+3.3, sp=0.4) | 50.0 (+6.7, sp=0.2) | 50.0 (+6.7, sp=0.4) |
| MBPP | 63.8 (+1.2, sp=0.4) | 63.2 (+0.0, sp=0.2) | 81.0 (−0.2, sp=0.4) |
| *MMLU-Pro* | | | |
| Chem. | 40.0 (−0.2, sp=0.4) | 53.2 (+2.2, sp=0.4) | 74.2 (+5.8, sp=0.4) |
| CS | 47.8 (+0.7, sp=0.2) | 50.7 (−2.7, sp=0.5) | 60.7 (+7.0, sp=0.6) |
| Law | 21.8 (−1.2, sp=0.6) | 13.0 (+0.2, sp=0.5) | 32.0 (+6.0, sp=0.6) |
| Phys. | 40.0 (−1.0, sp=0.4) | 51.2 (−2.4, sp=0.4) | 71.8 (+2.0, sp=0.4) |

## 4.2. Main Results

Figure 5 compares RLKV with baselines on four reasoning benchmarks (GSM8K, Math500, AIME24, MBPP) and four knowledge subsets of MMLU-Pro (Chemistry, Computer Science, Law, Physics) at sparsity levels of 0.2, 0.4, 0.6, and 0.8. Complete numerical results are provided in Appendix C. Overall, RLKV outperforms the baselines by up to 20%, and on some tasks even surpasses the full KV cache baseline.

**Reasoning Tasks.** RLKV consistently outperforms all methods across sparsity levels, with particularly strong advantages at high sparsity, such as 0.4 and 0.6, where baselines degrade substantially.

**Knowledge Tasks.** RLKV also maintains competitive accuracy on the four MMLU-Pro subsets across all sparsity levels. This suggests the effectiveness and robustness of our approach beyond the mathematical reasoning domain.

*Table 2.* **Long-context generalization on LongReason. (64K-input subset, 70K context)** Accuracy on Llama-3.1-8B-R1 and Qwen-3-4B-Thinking across four sparsity levels. H2O is omitted (out of memory). RLKV's adapters are trained with rollouts capped at 8K tokens and evaluated at 70K context length. Qwen-2.5-7B-R1 is excluded (full-attention accuracy: 0.5%).

| Method | sp = 0.2 | sp = 0.4 | sp = 0.6 | sp = 0.8 |
|---|---|---|---|---|
| *Llama-3.1-8B-R1* | | | | |
| Full | 49.25 | | | |
| R-KV | 0.0 (-49.25) | 0.0 (-49.25) | 0.0 (-49.25) | 0.0 (-49.25) |
| DuoAttention | 49.5 (+0.25) | 48.75 (-0.5) | 35.25 (-14.0) | 1.5 (-47.75) |
| KVZip | 48.0 (-1.25) | 49.0 (-0.25) | 36.0 (-13.25) | 4.75 (-44.5) |
| RLKV (Ours) | **50.5** (+1.25) | **52.5** (+3.25) | **45.25** (-4.0) | **15.0** (-34.25) |
| *Qwen-3-4B-Thinking* | | | | |
| Full | 70.25 | | | |
| R-KV | 1.25 (-69.0) | 0.75 (-69.5) | 0.5 (-69.75) | 0.5 (-69.75) |
| DuoAttention | 69.75 (-0.5) | 63.25 (-7.0) | 55.5 (-14.75) | 1.0 (-69.25) |
| KVZip | **71.25** (+1.0) | 67.25 (-3.0) | 53.0 (-17.25) | 0.25 (-70.0) |
| RLKV (Ours) | 68.5 (-1.75) | **68.25** (-2.0) | **58.25** (-12.0) | **3.0** (-67.25) |

**Near-Lossless Sparsity Thresholds.** Table 1 shows that RLKV achieves 20–60% KV cache reduction with near-lossless performance across various tasks and models.

**Long-context Generalization.** We evaluate on LongReason at four sparsity levels (Table 2). RLKV substantially outperforms all baselines on both models, with the gap widening at higher sparsity, suggesting that the heads it preserves capture reasoning behavior that transfers cleanly from 8K training to 70K inference. Head-reallocation baselines (DuoAttention, KVZip) produce more incorrect answers at high sparsity, while R-KV collapses into a repetitive loop.

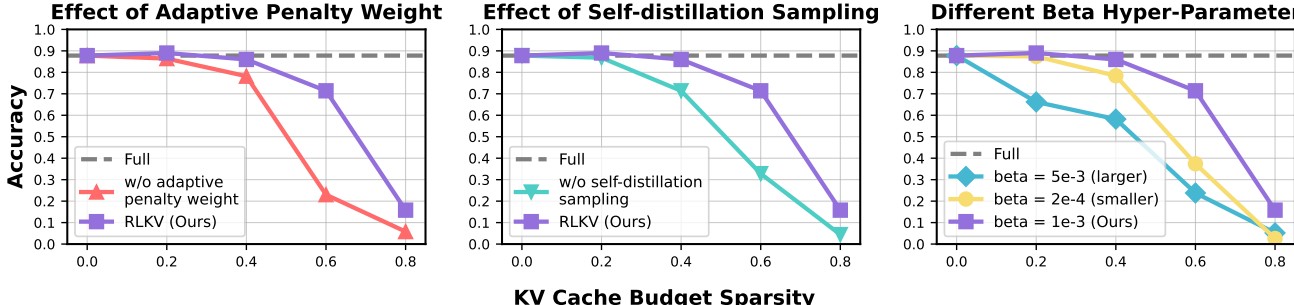

*Figure 6.* **Ablation of key components in RLKV training.** We conduct ablation studies on Qwen-2.5-7B-R1 with evaluation on Math500. *Left*: Adaptive penalty weighting stabilizes training under sparsity. *Middle*: Self-distillation sampling yields more stable reward signals than using overly difficult problems. *Right*: Base L1 penalty weight $\beta = 0.001$ provides the best sparsity–performance trade-off.

*Table 3.* **End-to-end speedup of RLKV on Math500 with continuous batching (SGLang).** Reported on Llama-3.1-8B-R1, a single A100 40G, with sink = 16 and local = 64. The first row is the full-attention baseline; all remaining rows are RLKV at the indicated sparsity.

| Sparsity | Concur. | Time (s) | Acc. (%) | Thpt. (tok/s) | Speedup | Theoretical |
|---|---|---|---|---|---|---|
| Full | 150 | 1,036 | 79.4 | 1,500 | 1.00× | – |
| 0.2 | 200 | 874 | 78.6 | 1,758 | **1.19×** | 1.24× |
| 0.4 | 250 | 666 | 79.6 | 2,185 | **1.56×** | 1.64× |
| 0.5 | 300 | 574 | 77.6 | 2,553 | **1.80×** | 1.95× |
| 0.6 | 375 | 504 | 73.8 | 2,941 | **2.06×** | 2.40× |

### 4.3. Inference Efficiency

We evaluate RLKV under realistic serving conditions by integrating it into SGLang (Zheng et al., 2024). The integration uses a dual KV pool: a standard paged cache for full-attention heads and a fixed-size circular buffer (sink + local) for compressed heads. The compressed pool requires only $\mathcal{O}(\max\_running\_requests \times W)$ tokens, where $W$ is the sink + local window, so memory freed from compressed heads can be rebalanced into the full pool, increasing the maximum number of concurrent requests. Table 3 shows that this translates into substantial end-to-end speedup, which grows with sparsity while remaining near-lossless at moderate sparsity. The measured speedups closely track the theoretical limit $1/((1 - s) + sW/L)$ derived from the residual cost of compressed heads, where $L$ is the average sequence length under continuous batching; the remaining gap reflects the dual-dispatch attention overhead, which custom fused kernels for heterogeneous-head attention would further close.

### 4.4. Ablation Studies

We conduct ablation studies on Math500 with Qwen-2.5-7B-R1 to evaluate three key training components of RLKV (adaptive penalty weighting, self-distillation sampling, base L1 penalty weight) and, on Llama-3.1-8B-R1, the effect of the inference-time sink+local window.

**Adaptive Penalty Weighting.** Figure 6 (left) demonstrates that adaptive penalty weighting significantly enhances performance by breaking the vicious cycle between

*Table 4.* **Effect of sink+local window on RLKV's compression frontier (Llama-3.1-8B-R1, Math500, SGLang).** Each cell is accuracy at the given (window, sparsity) with delta from the full-attention baseline (80.8); cells within 3 points of full are green.

| Sink + Local | sp = 0.2 | sp = 0.4 | sp = 0.6 | sp = 0.8 |
|---|---|---|---|---|
| Full | | 80.8 | | |
| 16 + 64 | 78.2 (-2.6) | 79.4 (-1.4) | 75.0 (-5.8) | 40.6 (-40.2) |
| 32 + 128 | 82.2 (+1.4) | 81.8 (+1.0) | 80.4 (-0.4) | 55.6 (-25.2) |
| 64 + 256 | 79.8 (-1.0) | 80.2 (-0.6) | 79.2 (-1.6) | 64.6 (-16.2) |

sparse rewards and dense L1 penalty. Without this mechanism, increasing adapter sparsity leads to degraded reasoning performance, which generates sparser reward signals while the L1 penalty remains dense, creating an imbalance that drives training toward collapse without recovery.

**Self-distillation Sampling.** Self-distillation sampling provides stable reward signals throughout training, as shown in Figure 6 (middle). In contrast to typical RLVR, training on problems suited to the model's reasoning capability maintains relatively stable reward signals throughout optimization, while training on challenging problems leads to unstable and sparse reward signals that provide weak and insufficient guidance for head identification.

**Base L1 penalty Weight.** The base regularization weight $\beta$ controls the strength of the L1 penalty applied to gating adapters during RL training. Figure 6 (right) shows that a moderate $\beta$ value of 0.001 achieves an optimal balance between sparsity and reward signal strength. Excessive penalty ($\beta = 0.005$) dominates the optimization process, weakening reward signals through over-compression, while insufficient penalty ($\beta = 0.0002$) fails to induce adequate sparsity, leading to premature convergence with limited exploration of the reward landscape.

**Sink + Local Window.** Table 4 ablates the compressed-head window on Llama-3.1-8B-R1 / Math500. At low sparsity, RLKV is robust to window choice; at high sparsity, larger windows extend the lossless range and soften the degradation cliff. Doubling the window costs only a marginal per-head KV residual but unlocks an additional

sparsity stop. In practice, we recommend modestly enlarging the window when targeting high sparsity.

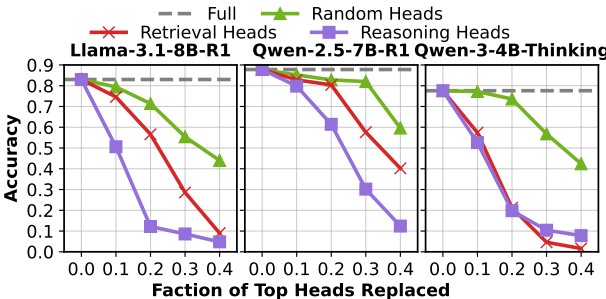

*Figure 7.* **Head sensitivity from high to low gating scores.** We rank heads by the learned gating scores and progressively replace the top fraction of heads with a compressed KV cache during inference, including randomly initialized heads, retrieval heads (DuoAttention), and reasoning-critical heads (RLKV).

## 5. Qualitative Analyses

In this section, we provide qualitative analyses to understand reasoning behaviors and reasoning models from a view of KV cache compression.

**Head sensitivity.** Figure 7 quantifies how sensitive the model is when compressing heads from high to low gating scores. For Llama-3.1-8B-R1 and Qwen-2.5-7B-R1, replacing reasoning-critical heads causes a significantly sharper drop than retrieval or random heads, confirming these heads are vital for maintaining reasoning behaviors. In Qwen-3-4B-Thinking, reasoning heads prove more impactful despite comparable sensitivity to retrieval heads; their preservation ensures better accuracy at high sparsity as shown in Figure 5. This indicates that the heads identified by RLKV are the primary drivers of reasoning performance.

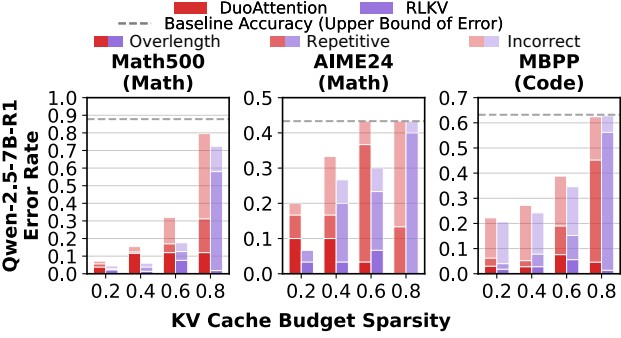

*Figure 8.* **Error modes induced by compressing different heads.** We categorize failures into repetitive, incorrect, and overlength errors, evaluated on instances that are solved correctly by the full KV cache baseline. See Appendix D for complete details.

**Error modes.** We categorize KV cache compression failures into three types: repetitive errors (repeating token sequences), incorrect errors (wrong final answers), and overlength errors (generation exceeding maximum context

length), as shown in Figure 12. We examine samples correctly solved by full KV cache but failed under compression. Compressing retrieval heads primarily causes overlength errors, where models maintain fluency but cannot reason efficiently within length constraints: validating our DuoAttention findings. Conversely, compressing reasoning-critical heads triggers repetitive and incorrect errors, showing these heads are essential for CoT consistency. Overlength errors rarely occur here, suggesting that preserved reasoning behavior maintains generation termination capability.

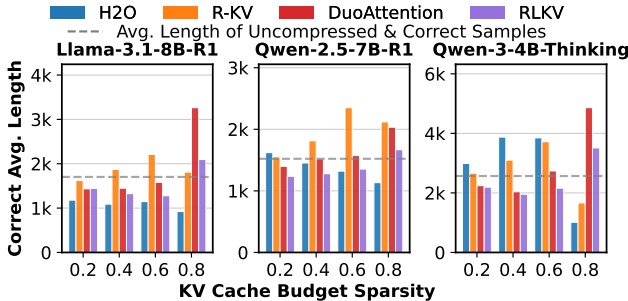

*Figure 9.* **Average response length of correct samples.** We report the average output length over samples that are answered correctly under both the full KV cache baseline and each compressed setting.

**Average response length.** We analyze the average response length on samples that remain correct under both the full KV cache baseline and each compressed setting (model, method and sparsity level). Figure 9 shows that token-dropping methods can appear to produce shorter outputs under aggressive compression, largely because they only solve easier instances. Although R-KV claims to preserve effective information, it often results in longer reasoning steps and poor performance. In contrast, DuoAttention often requires significantly longer reasoning steps to reach correct solutions, paying a higher computational cost for its retrieval-based allocation. Overall, RLKV maintains strong accuracy with competitive response lengths, suggesting a better trade-off between capability preservation and inference efficiency.

## 6. Conclusion

We propose RLKV, a novel reasoning-critical head identification method to guide KV cache compression in reasoning models that directly optimizes each head's cache usage against reasoning quality through reinforcement learning. RLKV achieves 20–60% KV cache reduction with near-lossless accuracy on reasoning and knowledge tasks, and up to $2.06\times$ end-to-end speedup under SGLang continuous batching. We further analyze the head sensitivity, error modes, and response-length cost of compression, revealing that reasoning-critical heads are the primary carriers of CoT consistency, empirically distinct from retrieval heads. RLKV provides a new perspective on understanding reasoning models and opens new avenues for introducing on-policy approaches to efficient methods.

## Acknowledgments

This paper is supported by Young Scientists Fund of the National Natural Science Foundation of China (NSFC) (No. 62506305), Zhejiang Leading Innovative and Entrepreneur Team Introduction Program (No. 2024R01007), Key Research and Development Program of Zhejiang Province (No. 2025C01026), Scientific Research Project of Westlake University (No. WU2025WF003), Chinese Association for Artificial Intelligence (CAAI) & Ant Group Research Fund – AGI Track (No. 2025CAAI-ANT-13). It is also supported by the research funds of the National Talent Program and Hangzhou Municipal Talent Program.

## Impact Statements

RLKV seeks to identify reasoning-critical heads to reveal inherent mechanisms underlying reasoning in reasoning large language models and use those heads to guide KV cache compression to improve the efficiency in reasoning decoding. By reducing memory usage and computational overhead during inference, this approach may contribute to more environmentally sustainable deployment of large-scale language models. As a KV cache compression technique, RLKV may introduce some degree of performance degradation, as is common for efficiency methods. Such effects can vary depending on downstream tasks and deployment settings, and should therefore be carefully evaluated when applied in real-world systems.

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

# A. Motivation Study

We provide a comprehensive motivation study on three mainstream reasoning models: Llama-3.1-8B-R1 (Guo et al., 2025), Qwen-2.5-7B-R1 (Guo et al., 2025), and Qwen-3-4B-Thinking [1] (Yang et al., 2025a), and their instruct variants: Llama-3.1-8B-Inst (Dubey et al., 2024), Qwen-2.5-7B-Inst [2] (Yang et al., 2024a), and Qwen-3-4B-Instruct [3](Yang et al., 2025a). We conduct the evaluation on two typical token-dropping methods: H₂O (Zhang et al., 2023) and R-KV (Cai et al., 2025), and one head-reallocation method: DuoAttention (Xiao et al., 2025), across four benchmarks, including GSM8K (Cobbe et al., 2021), Math500 (Lightman et al., 2023), AIME24 (Mathematical Association of America, 2024), MBPP (Austin et al., 2021). Figure 10 presents that all compression methods maintain relatively stable performance on instruct models but drop substantially on reasoning models as compression increases.

We further analyze the error modes on reasoning models in the above evaluation. We observed three error modes: repetitive errors (excessively repeating token sequences), incorrect errors (generating wrong answers), and overlength errors (generating sequences that exceed normal length baselines), as illustrated in Figure 12. The detailed error modes can be seen in Figure 11.

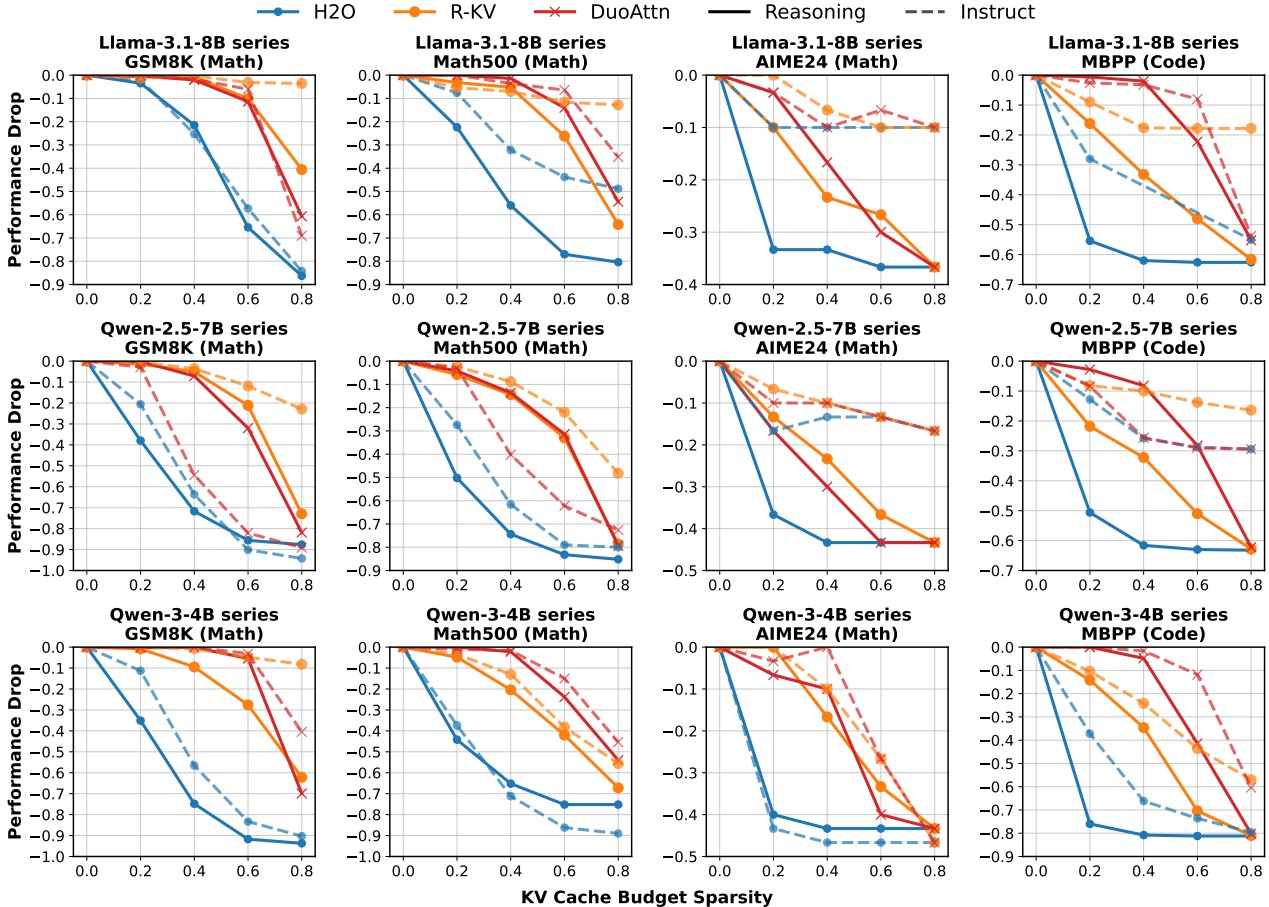

*Figure 10.* Comprehensive evaluation of KV cache compression methods across all model pairs and benchmarks reveals consistent patterns of performance degradation. H₂O, R-KV, and DuoAttention maintain relatively stable performance on instruction-following models but exhibit significant drops on their reasoning counterparts as the KV cache budget decreases. This performance degradation becomes particularly severe at higher sparsity levels, with notable declines observed on reasoning-intensive benchmarks including GSM8k, Math500, AIME24, and MBPP.

---

[1] It is the Qwen3-4B-Thinking-2507 instead of Qwen3-4B, which is a hybrid model in reasoning and instruct.

[2] We use Qwen-2.5-Math-7B-Instruct (Yang et al., 2024a) as the instruct baseline, abbreviated as Qwen-2.5-7B-Inst for naming consistency, since Qwen-2.5-7B-R1 (deepseek-ai/DeepSeek-R1-Distill-Qwen-7B) was based on Qwen-2.5-Math-7B

[3] It is the Qwen3-4B-Instruct-2507.

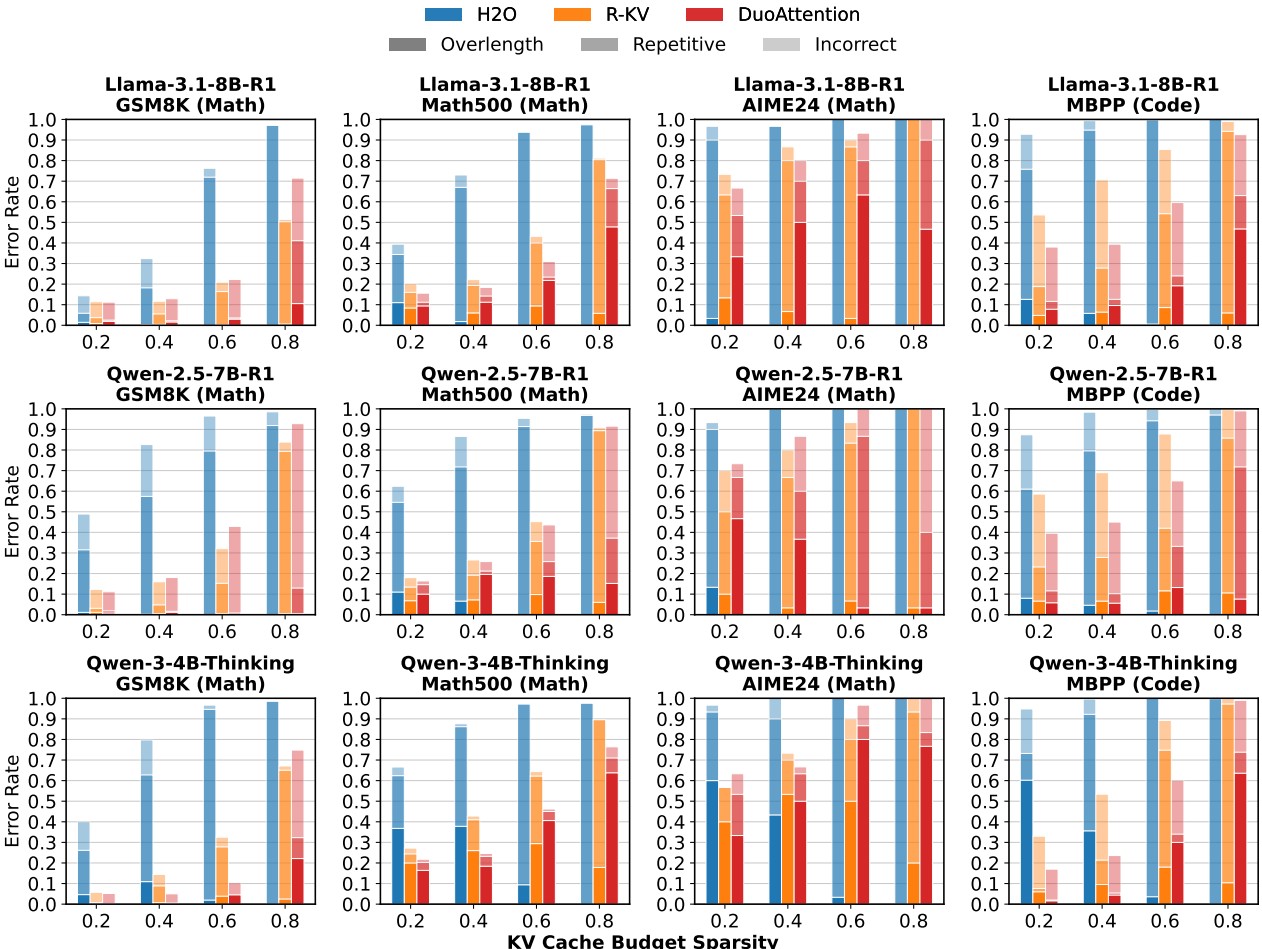

*Figure 11.* Comprehensive error mode analyses of KV cache compression methods across reasoning models reveal distinct failure patterns. Token-dropping methods (H$_2$O, R-KV) consistently exhibit repetitive errors, as they inevitably discard reasoning-critical information during compression. In contrast, the head-reallocation method DuoAttention tends to show more over-length errors compared to token-dropping methods, suggesting that while it relatively preserves sequence information integrity, it still struggles to fully preserve reasoning capability.

## B. Experiment Details

**Dataset Construction.** We construct training data from the DeepScaleR dataset (Luo et al., 2025), which contains about 40,000 diverse and challenging mathematical reasoning problems. For each model, we generate solutions using the respective reasoning model with greedy decoding, filter correct solutions, then randomly sample 3,000 problems for training. The selected problems are distributed across different output token lengths as follows: 600 problems each for 0-2k and 2k-4k tokens, 1,000 problems for 4k-6k tokens, and 800 problems for 6k-8k tokens.

**Hardware and Hyperparameter Settings.** All trainings are conducted on 2 NVIDIA A100 GPUs (80GB) for several hours, one for backward computation and one for sample generation. Training runs for 2 epochs, totaling 185 steps with a batch size of 32. All evaluations are conducted on NVIDIA RTX5090 GPUs. We optimize the gating adapters using AdamW optimizer with $\beta_1 = 0.9$, $\beta_2 = 0.999$, weight decay of 0.017, and learning rate of 0.01 with constant schedule. For GRPO training configuration, we disable KL penalty and use recommendation setting of AReaL; for GRPO sampling configuration, we use 4 samples per query with sampling temperature of 1.0. The hyperparameters are shown in Table Table 5.

**Local Attention Implementation.** During training, we employ an efficient block-sparse attention approximation implementation (Guo et al., 2024) in the FSDP engine of AReaL (Fu et al., 2025) to update adapter weights, while using mask matrices for prefilling and custom Triton kernels for decoding in SGLang (Zheng et al., 2024) to generate samples. For

**Repetitive Error:** Okay, so I have this problem here where I need to find the greatest four-digit number N such that if you change any of its digits to 1, the resulting number is divisible by 7. Then, I have to find Q + R where Q is the quotient and R is the remainder when N is divided by 1000. ...(after about 2000 tokens) but that's not necessarily the case because 1111 is 7*11*13, which is 7*143, but 1111 is 7*11*13, so 1111 is 7*11*13, which is 7*11*13, which is 7*11*13, which is 7*11*13, which is 7*11*13 ...(keep repeating)

**Incorrect Error:** Okay, so I have this problem about residents in Aimeville and the things they own. Let me try to figure it out step by step. Hmm, it's a problem involving sets and maybe using some principles from set theory or combinatorics. I remember something about inclusion-exclusion principles from my math classes, so maybe that's what I need here. ... (after about 6500 tokens) Therefore, the number of residents who own all four things is \\(\\boxed{219}\\). #but correct answer is 73

**Overlength Error:** Okay, so I have this probability problem here: there are four points, A, B, C, and D, ... In this diagram, the green edges represent the labeling where $AB$ and $CD$ intersect, and the blue and red edges represent the equally likely labelings where $AB$ and $CD$ do not intersect #stop at 8k maximum output length

*Figure 12.* The instances of three error modes, including repetitive errors (excessively repeating token sequences), incorrect errors (wrong final answers), and overlength errors (generation exceeding the maximum context length).

*Table 5.* Training Hyperparameters.

| Parameter | Llama-3.1-8B-R1 | Qwen-2.5-7B-R1 | Qwen-3-4B-Thinking |
|---|---|---|---|
| Regularization weight $\beta$ | 1e-3 | 1e-3 | 2.5e-3 |
| Reward threshold $\tau$ | 0.5 | 0.55 | 0.5 |
| Top_P | 1.0 | 1.0 | 0.95 |
| Sink token size | 128 | 128 | 128 |
| Local token size | 256 | 256 | 256 |
| Max sequence length | 8192 | 8192 | 8192 |

inference, we only store the full KV cache for reasoning-critical heads, while others only maintain the partial KV cache of the first 16 sink tokens and the recent 64 local tokens.

**SGLang Engine Limitations.** Our current dual-pool integration in SGLang v0.5.2 does not yet support radix-cache reuse or paged attention for the compressed-head buffer. Since both the RLKV runs and the full-attention baseline operate under the same configuration, this does not affect the relative speedup reported in Table 3; supporting these SGLang features would make RLKV more practical for production serving. In addition, SGLang generation is not strictly deterministic across runs, which introduces minor variance in absolute accuracy; this variance is an artifact of the engine and applies equally to the full-attention baseline.

**Baseline Implementation.** To ensure fair comparison with baseline methods, we make several adjustments. For H$_2$O and R-KV, we augment them with the same sink and local token overhead (16+64 tokens) that our method uses. Since H$_2$O and R-KV only support preset fixed KV cache budgets, we convert their fixed budgets to dynamic allocation that increases with sequence length. For example, if the fixed budget is 50% of the full KV cache, then at sequence length 1000, they use 500 tokens of KV cache, and at sequence length 2000, they use 1000 tokens of KV cache. For DuoAttention, we replicate their approach with default settings on our models and use the same inference settings as our method.

**Training Cost.** The training of the adapters is computationally modest: on 2 A100 GPUs, our method consumes 40, 22, and 36 GPU-hours for Llama-3.1-8B-R1, Qwen-2.5-7B-R1, and Qwen-3-4B-Thinking, respectively.

**Evaluation Settings.** We evaluate all methods using greedy decoding on RTX 5090 36G GPUs or RTX 4090 24G GPUs with batch size of 1. For all datasets, we use regex to extract the final answer from the generated text, using Pass@1 as the evaluation metric. For GSM8K, Math500, MBPP and MMLU-Pro subsets, we use 8192 max sequence length; for AIME24, we use 16384 max sequence length. We achieved near official reported performance without KV cache compression. We use eager attention implementation for H$_2$O and R-KV since they need to use attention scores, while we use flash attention for DuoAttention and our method.

**Inference Engines.** The accuracy results reported in Table 1 are produced with two inference engines: the math and code benchmarks (GSM8K, Math500, AIME24, MBPP) are evaluated with HuggingFace Transformers, while the MMLU-Pro subsets are evaluated with our SGLang integration. The engine choice does not bias the comparison: within each benchmark,

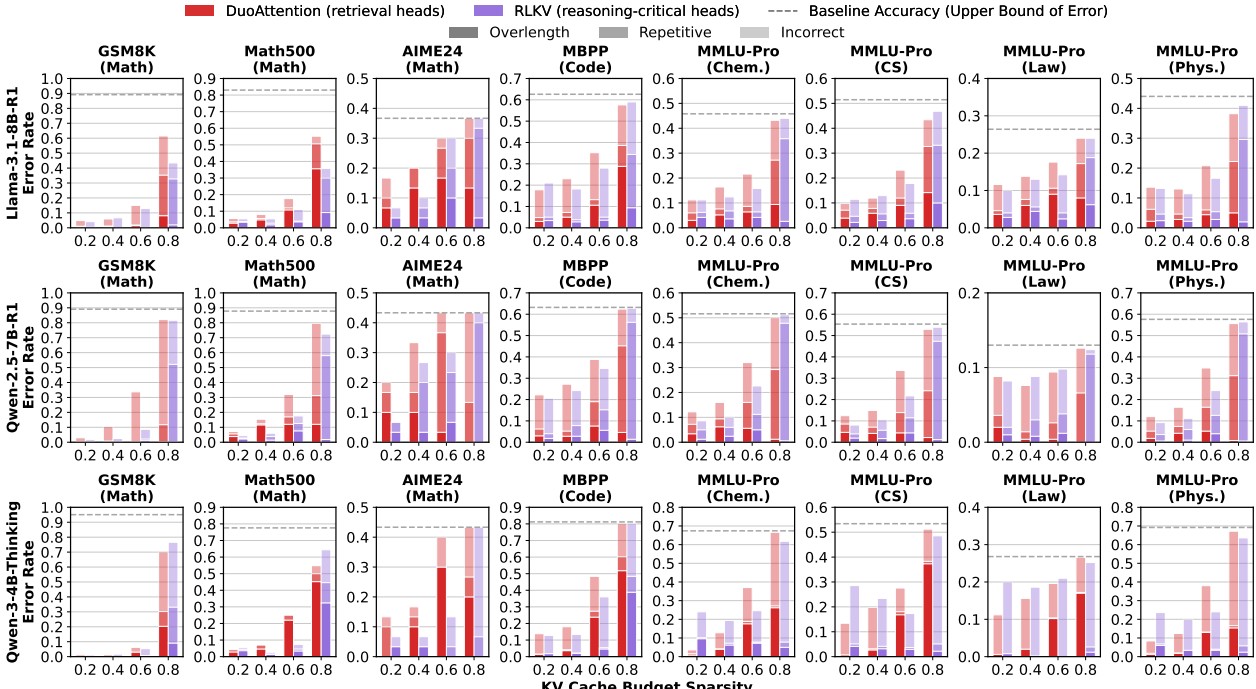

*Figure 13.* **Error modes induced by compressing different heads.** We categorize failures into repetitive, incorrect, and overlength errors, evaluated on instances that are solved correctly by the full KV cache baseline.

all methods (including the full-attention baseline used for the delta) run under the same engine, so the per-cell delta isolates the effect of compression rather than the engine.

**Prompt Template.** We follow the prompt setting recommended by DeepSeek-R1 (Guo et al., 2025) in both training and evaluation without additional prompt engineering. For example, we use the following template in math problems:

```
Solve the following math problem efficiently and clearly.  The last line
of your response should be of the following format:  'Therefore, the
final answer is:  $\\boxed{ANSWER}$.  I hope it is correct' (without
quotes) where ANSWER is just the final number or expression that solves
the problem.  Think step by step before answering.

QUESTION
```

## C. Complete numerical results

Table 6, Table 7 and Table 8 present the complete numerical results of RLKV and baselines for Llama-3.1-8B-R1, Qwen-2.5-7B-R1, and Qwen-3-4B-Thinking respectively, across all benchmarks and KV cache compression budgets. Values in parentheses indicate the performance difference compared to the full KV cache setting, with positive values in green indicating improvement and negative values in red indicating degradation.

## D. Details of Error Modes Analyses

Figure 13 presents the comprehensive error mode analyses across all models and benchmarks. Our findings in Figure 8 is consistent with our observations in the main experiments, except for the evaluation of Qwen-3-4B-Thinking on Math500 and MBPP at 0.8 sparsity.

**Tokens per Sample Distribution with Full KV Cache**

*Figure 14.* The distribution of output lengths on Math500 and AIME24 benchmarks with Llama-3.1-8B-R1, Qwen-2.5-7B-R1, and Qwen-3-4B-Thinking models with full KV cache.

## E. An Implicitly Unfair Comparison in Fixed-Budget Evaluation

This section discusses the motivation for using a dynamic budget instead of a fixed budget for KV cache compression evaluation. Existing long-context compression works (Li et al., 2024; Yang et al., 2024b; Qin et al., 2025; Fu et al., 2024; Tang et al., 2025; Xiao et al., 2025; Bhaskar et al., 2025) typically evaluate on in-context recall tasks, where each sample's prompt length is fixed/controlled. A fixed budget of the form budget = sparsity × prompt_length then yields a roughly consistent compression ratio per sample, so fixed budgets are fair in that setting.

For reasoning tasks, however, the response length is often much larger than the prompt, as shown in Figure 14. If we use a global fixed budget (e.g., 1k tokens), any sample whose full output fits within 1k tokens is uncompressed, while longer samples are compressed. Thus, different samples experience very different compression ratios, and fixed budgets are not fair at the per-sample level.

In R-KV (Cai et al., 2025), the reported compression rate is computed as budget/average_full_length. For example, R-KV achieves the compression ratio of 66.2% for Math500 on Llama-3.1-8B-R1, with a fixed budget of 200 and an average full length of 3019. However, a large fraction of samples are uncompressed and thus produce the same responses as the full model. This makes the reported compression ratio optimistic.

## F. Comparison of Fixed Budget and Dynamic Budget for R-KV and H2O

In our evaluations, we adopt a dynamic budget strategy where each sample's budget is determined by its full length multiplied by the target sparsity, to ensure consistent compression ratios across samples. To illustrate the impact of this choice, we compare the performance of R-KV and H2O under both fixed and dynamic budget settings on Llama-3.1-8B-R1,

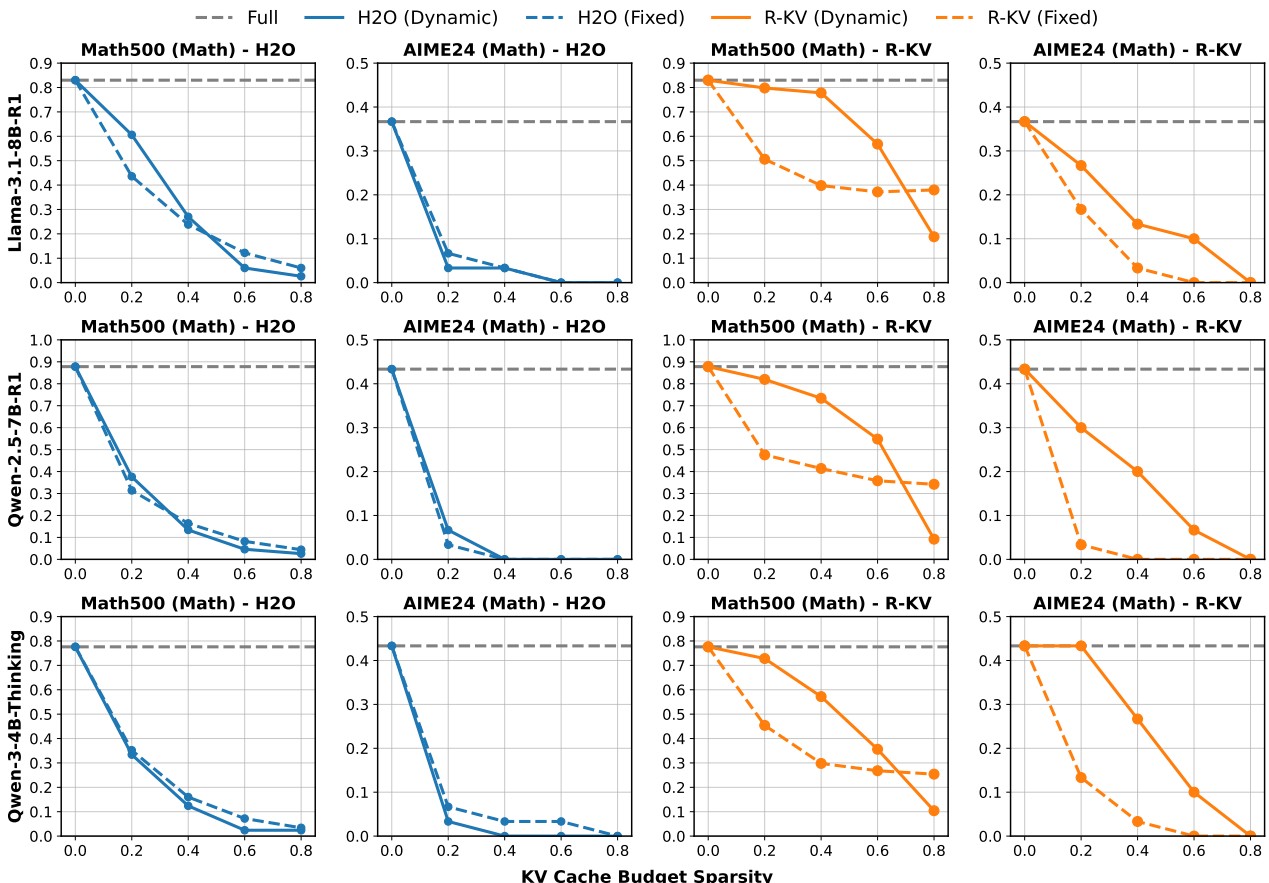

*Figure 15.* Performance comparison of R-KV and H2O under fixed budget and dynamic budget settings on Llama-3.1-8B-R1, Qwen-2.5-7B-R1, and Qwen-3-4B-Thinking across Math500 and AIME24 at sparsity levels of 0.2, 0.4, 0.6, and 0.8. The fixed-budget R-KV performs significantly worse than the dynamic-budget variant at 0.2, 0.4, and 0.6 sparsity, and only becomes better at 0.8 sparsity, while H2O maintains similar performance across both settings.

Qwen-2.5-7B-R1, and Qwen-3-4B-Thinking across Math500 and AIME24 at sparsity levels of 0.2, 0.4, 0.6, and 0.8. In this comparison, the fixed budget per-sample is estimated as budget(sample) = sparsity × full_length(sample), where full_length(sample) is the length of the response generated by the full KV cache model for that specific sample.

As shown in Figure 15, fixed-budget R-KV performs significantly worse than our dynamic-budget variant at 0.2–0.6 sparsity, and only surpasses at 0.8 sparsity, while H2O maintains similar performance. This shows that our modification does not weaken the baselines; instead, it corrects an overly optimistic compression estimate and yields a more faithful comparison.

## G. Limitations and Future Work

First, the distribution of reasoning-critical heads varies non-trivially across models and tasks (Figure 3), reflecting the heterogeneous nature of reasoning mechanisms; the current approach captures this as a static artifact without decomposing it into finer functional categories. Second, RLKV produces static gating scores fixed at training time, while query-adaptive dynamic gating would be more flexible – yet how to balance the overhead of dynamic decisions against the efficiency gains from compression remains an open question. Third, RLKV suffers substantial performance degradation at extremely high compression ratios (sparsity $\geq 0.8$), and pushing compression further without quality loss likely requires combining head-reallocation with orthogonal directions such as KV cache quantization.

RLKV's on-policy formulation generalizes beyond KV cache compression. First, the same paradigm extends to other efficiency methods such as pruning, sparse attention, and token compression (Ma et al., 2023; Zhu et al., 2026; Feng et al., 2024; Gao et al., 2024; Tao et al., 2025a; Mei et al., 2026). Second, it extends across tasks, to video (Tao et al., 2025a; Shao

et al., 2025; Wang et al., 2026a), streaming (Chen et al., 2026; Wang et al., 2026b), video-audio understanding (Tao et al., 2026), and vision-language-action models (Yang et al., 2025c). Third, applying these efficiency methods to multimodal reasoning (Dong et al., 2025; 2026) and agentic (Wen et al., 2025; Tian et al., 2025; Liu et al., 2025a) reasoning is a natural next step, where rollout rewards remain the only reliable signal of how compression interacts with multi-step reasoning, tool use, and planning.

*Table 6.* Llama-3.1-8B-R1 performance (%) under different KV cache compression methods and budgets. RLKV (**Ours**) shows competitive performance across settings. Red background denotes performance below the full-KV-cache baseline, whereas green background denotes performance above it. For all values, higher is better. The best result of the metric in each benchmark is in **bold**.

| Dataset | Method | KV Cache Budget Sparsity | | | |
| | | 0.2 | 0.4 | 0.6 | 0.8 |
|---|---|---|---|---|---|
| GSM8K (Math) | H2O | 85.7 (-3.5) | 67.6 (-21.5) | 23.7 (-65.4) | 2.9 (-86.3) |
| | R-KV | 88.5 (-0.6) | **88.3** (-0.8) | 79.1 (-10.1) | **48.6** (-40.6) |
| | DuoAttention | 88.8 (-0.4) | 87.1 (-2.0) | 77.8 (-11.4) | 28.5 (-60.6) |
| | KVZip | **89.2** (+0.1) | 81.8 (-7.4) | 78.3 (-10.8) | 34.3 (-54.9) |
| | RLKV (Ours) | **89.2** (+0.1) | 86.8 (-2.3) | **79.5** (-9.7) | 47.4 (-41.8) |
| Math500 (Math) | H2O | 60.6 (-22.4) | 27.0 (-56.0) | 6.0 (-77.0) | 2.6 (-80.4) |
| | R-KV | 79.8 (-3.2) | 77.8 (-5.2) | 56.8 (-26.2) | 18.8 (-64.2) |
| | DuoAttention | **84.4** (+1.4) | 81.6 (-1.4) | 69.0 (-14.0) | 28.6 (-54.4) |
| | KVZip | 79.0 (-4.0) | 75.4 (-7.6) | 65.4 (-17.6) | 24.4 (-58.6) |
| | RLKV (Ours) | **84.4** (+1.4) | **84.6** (+1.6) | **78.0** (-5.0) | **49.6** (-33.4) |
| AIME24 (Math) | H2O | 3.3 (-33.3) | 3.3 (-33.3) | 0.0 (-36.7) | 0.0 (-36.7) |
| | R-KV | 26.7 (-10.0) | 13.3 (-23.3) | 10.0 (-26.7) | 0.0 (-36.7) |
| | DuoAttention | 33.3 (-3.3) | 20.0 (-16.7) | 6.7 (-30.0) | 0.0 (-36.7) |
| | KVZip | 30.0 (-6.7) | 20.0 (-16.7) | 13.3 (-23.3) | **3.3** (-33.3) |
| | RLKV (Ours) | **36.7** (+0.0) | **40.0** (+3.3) | **20.0** (-16.7) | 0.0 (-36.7) |
| MBPP (Code) | H2O | 7.2 (-55.4) | 0.6 (-62.0) | 0.0 (-62.6) | 0.0 (-62.6) |
| | R-KV | 46.4 (-16.2) | 29.4 (-33.2) | 14.6 (-48.0) | 1.0 (-61.6) |
| | DuoAttention | 62.0 (-0.6) | 60.6 (-2.0) | 40.4 (-22.2) | 7.4 (-55.2) |
| | KVZip | 59.8 (-2.8) | 57.8 (-4.8) | 39.6 (-23.0) | **12.6** (-50.0) |
| | RLKV (Ours) | **62.8** (+0.2) | **63.8** (+1.2) | **51.8** (-10.8) | 6.0 (-56.6) |
| MMLU-Pro (Chem.) | H2O | 26.8 (-19.0) | 7.8 (-38.0) | 1.6 (-44.2) | 0.0 (-45.8) |
| | R-KV | 41.0 (-4.8) | 31.2 (-14.6) | 21.6 (-24.2) | 3.4 (-42.4) |
| | DuoAttention | 43.2 (-2.6) | 36.4 (-9.4) | 30.4 (-15.4) | **3.8** (-42.0) |
| | KVZip | 40.6 (-5.2) | 38.6 (-7.2) | 0.2 (-45.6) | 0.0 (-45.8) |
| | RLKV (Ours) | **44.4** (-1.4) | **44.8** (-1.0) | **37.8** (-8.0) | 2.2 (-43.6) |
| MMLU-Pro (CS) | H2O | 42.0 (-9.5) | 27.6 (-23.9) | 6.6 (-44.9) | 0.0 (-51.5) |
| | R-KV | 47.6 (-3.9) | 44.4 (-7.1) | 31.2 (-20.2) | 9.5 (-42.0) |
| | DuoAttention | 48.3 (-3.2) | 46.1 (-5.4) | 34.4 (-17.1) | **9.8** (-41.7) |
| | KVZip | 47.8 (-3.7) | 40.5 (-11.0) | 2.9 (-48.5) | 0.2 (-51.2) |
| | RLKV (Ours) | **49.3** (-2.2) | **47.6** (-3.9) | **43.2** (-8.3) | 5.1 (-46.3) |
| MMLU-Pro (Law) | H2O | 21.2 (-5.2) | 16.2 (-10.2) | 3.8 (-22.6) | 0.0 (-26.4) |
| | R-KV | 22.0 (-4.4) | 15.8 (-10.6) | 13.8 (-12.6) | 1.2 (-25.2) |
| | DuoAttention | **24.2** (-2.2) | 19.8 (-6.6) | 13.0 (-13.4) | 3.6 (-22.8) |
| | KVZip | 16.6 (-9.8) | 15.6 (-10.8) | 0.8 (-25.6) | 0.8 (-25.6) |
| | RLKV (Ours) | 23.6 (-2.8) | **22.8** (-3.6) | **20.6** (-5.8) | **4.2** (-22.2) |
| MMLU-Pro (Phys.) | H2O | 27.0 (-17.0) | 13.0 (-31.0) | 3.6 (-40.4) | 0.0 (-44.0) |
| | R-KV | 38.8 (-5.2) | 36.4 (-7.6) | 26.8 (-17.2) | 5.4 (-38.6) |
| | DuoAttention | 40.6 (-3.4) | 38.6 (-5.4) | 29.4 (-14.6) | **6.6** (-37.4) |
| | KVZip | 38.4 (-5.6) | 36.2 (-7.8) | 1.2 (-42.8) | 0.0 (-44.0) |
| | RLKV (Ours) | **40.8** (-3.2) | **42.8** (-1.2) | **37.4** (-6.6) | 4.4 (-39.6) |

*Table 7.* Qwen-2.5-7B-R1 performance (%) under different KV cache compression methods and budgets. RLKV (**Ours**) shows competitive performance across settings. Red background denotes performance below the full-KV-cache baseline, whereas green background denotes performance above it. For all values, higher is better. The best result of the metric in each benchmark is in **bold**.

| Dataset | Method | KV Cache Budget Sparsity | | | |
| | | 0.2 | 0.4 | 0.6 | 0.8 |
|---|---|---|---|---|---|
| GSM8K (Math) | H2O | 51.1 (-38.0) | 17.4 (-71.7) | 3.5 (-85.6) | 1.4 (-87.6) |
| | R-KV | 87.7 (-1.4) | 84.0 (-5.1) | 67.9 (-21.1) | **16.1** (-72.9) |
| | DuoAttention | 88.9 (-0.2) | 82.0 (-7.1) | 57.1 (-32.0) | 7.2 (-81.9) |
| | KVZip | 88.2 (-0.8) | 85.0 (-4.1) | 66.7 (-22.4) | 2.8 (-86.3) |
| | RLKV (Ours) | **90.7** (+1.6) | **90.1** (+1.0) | **83.1** (-6.0) | 7.7 (-81.3) |
| Math500 (Math) | H2O | 37.6 (-50.2) | 13.4 (-74.4) | 4.6 (-83.2) | 2.6 (-85.2) |
| | R-KV | 82.0 (-5.8) | 73.4 (-14.4) | 54.8 (-33.0) | 9.2 (-78.6) |
| | DuoAttention | 83.6 (-4.2) | 74.2 (-13.6) | 56.4 (-31.4) | 8.4 (-79.4) |
| | KVZip | 87.0 (-0.8) | 81.6 (-6.2) | 60.8 (-27.0) | 3.4 (-84.4) |
| | RLKV (Ours) | **89.0** (+1.2) | **86.0** (-1.8) | **71.4** (-16.4) | **15.8** (-72.0) |
| AIME24 (Math) | H2O | 6.7 (-36.7) | 0.0 (-43.3) | 0.0 (-43.3) | **0.0** (-43.3) |
| | R-KV | 30.0 (-13.3) | 20.0 (-23.3) | 6.7 (-36.7) | **0.0** (-43.3) |
| | DuoAttention | 26.7 (-16.7) | 13.3 (-30.0) | 0.0 (-43.3) | **0.0** (-43.3) |
| | KVZip | 43.3 (+0.0) | **26.7** (-16.7) | 6.7 (-36.7) | **0.0** (-43.3) |
| | RLKV (Ours) | **50.0** (+6.7) | **26.7** (-16.7) | **13.3** (-30.0) | **0.0** (-43.3) |
| MBPP (Code) | H2O | 12.6 (-50.6) | 1.6 (-61.6) | 0.2 (-63.0) | 0.0 (-63.2) |
| | R-KV | 41.4 (-21.8) | 31.0 (-32.2) | 12.2 (-51.0) | 0.4 (-62.8) |
| | DuoAttention | 60.4 (-2.8) | 55.0 (-8.2) | 35.0 (-28.2) | **1.0** (-62.2) |
| | KVZip | 56.6 (-6.6) | 50.0 (-13.2) | 39.2 (-24.0) | 0.4 (-62.8) |
| | RLKV (Ours) | **63.2** (+0.0) | **56.8** (-6.4) | **40.2** (-23.0) | 0.6 (-62.6) |
| MMLU-Pro (Chem.) | H2O | 18.8 (-32.8) | 7.2 (-44.4) | 3.0 (-48.6) | 0.4 (-51.2) |
| | R-KV | 43.4 (-8.2) | 32.8 (-18.8) | 17.4 (-34.2) | 1.0 (-50.6) |
| | DuoAttention | 47.4 (-4.2) | 44.8 (-6.8) | 24.8 (-26.8) | **2.2** (-49.4) |
| | KVZip | 51.0 (-0.6) | 41.2 (-10.4) | 22.8 (-28.8) | 0.4 (-51.2) |
| | RLKV (Ours) | **55.0** (+3.4) | **50.8** (-0.8) | **36.0** (-15.6) | 0.4 (-51.2) |
| MMLU-Pro (CS) | H2O | 39.5 (-15.9) | 22.7 (-32.7) | 11.2 (-44.1) | 2.9 (-52.4) |
| | R-KV | 44.6 (-10.7) | 39.3 (-16.1) | 27.3 (-28.0) | 1.9 (-53.4) |
| | DuoAttention | 53.7 (-1.7) | 49.3 (-6.1) | 27.3 (-28.0) | **4.2** (-51.2) |
| | KVZip | 51.7 (-3.7) | 49.0 (-6.3) | 20.0 (-35.4) | 1.0 (-54.4) |
| | RLKV (Ours) | **56.8** (+1.5) | **54.6** (-0.7) | **43.7** (-11.7) | 1.7 (-53.7) |
| MMLU-Pro (Law) | H2O | **13.8** (+0.8) | 11.4 (-1.6) | 3.0 (-10.0) | 0.0 (-13.0) |
| | R-KV | 12.0 (-1.0) | 7.6 (-5.4) | 3.8 (-9.2) | 0.2 (-12.8) |
| | DuoAttention | 9.8 (-3.2) | **15.6** (+2.6) | **10.6** (-2.4) | **2.4** (-10.6) |
| | KVZip | 10.8 (-2.2) | 9.0 (-4.0) | 3.8 (-9.2) | 0.4 (-12.6) |
| | RLKV (Ours) | 11.8 (-1.2) | 12.6 (-0.4) | 9.8 (-3.2) | **2.4** (-10.6) |
| MMLU-Pro (Phys.) | H2O | 24.8 (-32.8) | 11.4 (-46.2) | 5.6 (-52.0) | 1.4 (-56.2) |
| | R-KV | 49.2 (-8.4) | 38.2 (-19.4) | 23.4 (-34.2) | 2.2 (-55.4) |
| | DuoAttention | 52.4 (-5.2) | 47.6 (-10.0) | 26.0 (-31.6) | **3.0** (-54.6) |
| | KVZip | 51.0 (-6.6) | 44.6 (-13.0) | 19.4 (-38.2) | 0.4 (-57.2) |
| | RLKV (Ours) | **56.8** (-0.8) | **53.8** (-3.8) | **37.0** (-20.6) | 1.2 (-56.4) |

*Table 8.* Qwen-3-4B-Thinking performance (%) under different KV cache compression methods and budgets. RLKV (**Ours**) shows competitive performance across settings. Red background denotes performance below the full-KV-cache baseline, whereas green background denotes performance above it. For all values, higher is better. The best result of the metric in each benchmark is in **bold**.

| Dataset | Method | KV Cache Budget Sparsity | | | |
|---|---|---|---|---|---|
| | | 0.2 | 0.4 | 0.6 | 0.8 |
| GSM8K (Math) | H2O | 60.0 (-35.1) | 20.2 (-74.8) | 3.3 (-91.7) | 1.4 (-93.7) |
| | R-KV | 94.2 (-0.8) | 85.6 (-9.5) | 67.5 (-27.6) | **32.9** (-62.2) |
| | DuoAttention | 94.8 (-0.2) | 95.0 (-0.1) | 89.6 (-5.5) | 25.2 (-69.9) |
| | KVZip | **95.2** (+0.2) | **95.1** (+0.0) | **91.4** (-3.6) | 0.4 (-94.7) |
| | RLKV (Ours) | 94.9 (-0.1) | 94.4 (-0.7) | 90.8 (-4.2) | 18.7 (-76.3) |
| Math500 (Math) | H2O | 33.4 (-44.2) | 12.4 (-65.2) | 2.4 (-75.2) | 2.4 (-75.2) |
| | R-KV | 72.8 (-4.8) | 57.2 (-20.4) | 35.6 (-42.0) | 10.4 (-67.2) |
| | DuoAttention | 78.2 (+0.6) | 75.4 (-2.2) | 53.8 (-23.8) | **23.6** (-54.0) |
| | KVZip | **79.2** (+1.6) | 83.0 (+5.4) | 71.2 (-6.4) | 1.0 (-76.6) |
| | RLKV (Ours) | 77.8 (+0.2) | **83.2** (+5.6) | **75.6** (-2.0) | 13.2 (-64.4) |
| AIME24 (Math) | H2O | 3.3 (-40.0) | 0.0 (-43.3) | 0.0 (-43.3) | **0.0** (-43.3) |
| | R-KV | 43.3 (+0.0) | 26.7 (-16.7) | 10.0 (-33.3) | **0.0** (-43.3) |
| | DuoAttention | 36.7 (-6.7) | 33.3 (-10.0) | 3.3 (-40.0) | **0.0** (-43.3) |
| | KVZip | **46.7** (+3.3) | 40.0 (-3.3) | 20.0 (-23.3) | **0.0** (-43.3) |
| | RLKV (Ours) | **46.7** (+3.3) | **50.0** (+6.7) | **30.0** (-13.3) | **0.0** (-43.3) |
| MBPP (Code) | H2O | 5.2 (-76.0) | 0.4 (-80.8) | 0.0 (-81.2) | 0.0 (-81.2) |
| | R-KV | 67.0 (-14.2) | 46.6 (-34.6) | 10.8 (-70.4) | 0.2 (-81.0) |
| | DuoAttention | **83.0** (+1.8) | 76.4 (-4.8) | 39.8 (-41.4) | **1.0** (-80.2) |
| | KVZip | 78.2 (-3.0) | 74.4 (-6.8) | **63.8** (-17.4) | 0.4 (-80.8) |
| | RLKV (Ours) | 82.4 (+1.2) | **81.0** (-0.2) | 55.2 (-26.0) | **1.0** (-80.2) |
| MMLU-Pro (Chem.) | H2O | 21.6 (-45.8) | 9.0 (-58.4) | 2.8 (-64.6) | 0.0 (-67.4) |
| | R-KV | 65.0 (-2.4) | 49.8 (-17.6) | 19.8 (-47.6) | 0.8 (-66.6) |
| | DuoAttention | **72.8** (+5.4) | 62.4 (-5.0) | 32.8 (-34.6) | 1.4 (-66.0) |
| | KVZip | 69.0 (+1.6) | 70.8 (+3.4) | 56.0 (-11.4) | 0.2 (-67.2) |
| | RLKV (Ours) | 64.6 (-2.8) | **72.2** (+4.8) | **62.6** (-4.8) | **7.4** (-60.0) |
| MMLU-Pro (CS) | H2O | 39.8 (-13.6) | 21.9 (-31.5) | 10.7 (-42.7) | 0.0 (-53.4) |
| | R-KV | 50.5 (-2.9) | 48.0 (-5.4) | 31.0 (-22.4) | 3.2 (-50.2) |
| | DuoAttention | 50.2 (-3.2) | 40.0 (-13.4) | 32.7 (-20.7) | 2.9 (-50.5) |
| | KVZip | **51.2** (-2.2) | **59.0** (+5.6) | 47.8 (-5.6) | 0.2 (-53.2) |
| | RLKV (Ours) | 47.8 (-5.6) | 53.2 (-0.2) | **63.2** (+9.8) | **10.5** (-42.9) |
| MMLU-Pro (Law) | H2O | 25.8 (-1.0) | 19.0 (-7.8) | 9.0 (-17.8) | 0.0 (-26.8) |
| | R-KV | 26.2 (-0.6) | 22.0 (-4.8) | 9.6 (-17.2) | 0.8 (-26.0) |
| | DuoAttention | 26.2 (-0.6) | 17.8 (-9.0) | 13.8 (-13.0) | 1.8 (-25.0) |
| | KVZip | 26.8 (+0.0) | **30.8** (+4.0) | 23.4 (-3.4) | 0.2 (-26.6) |
| | RLKV (Ours) | **27.0** (+0.2) | 27.2 (+0.4) | **27.6** (+0.8) | **7.4** (-19.4) |
| MMLU-Pro (Phys.) | H2O | 25.6 (-43.6) | 13.2 (-56.0) | 4.6 (-64.6) | 0.2 (-69.0) |
| | R-KV | 61.6 (-7.6) | 50.4 (-18.8) | 24.2 (-45.0) | 1.6 (-67.6) |
| | DuoAttention | 66.6 (-2.6) | 62.8 (-6.4) | 34.6 (-34.6) | 2.2 (-67.0) |
| | KVZip | 66.6 (-2.6) | **70.4** (+1.2) | 53.4 (-15.8) | 0.0 (-69.2) |
| | RLKV (Ours) | **66.8** (-2.4) | 69.8 (+0.6) | **64.2** (-5.0) | **8.4** (-60.8) |

