# OpenReview forum: "Which Heads Matter for Reasoning? RL-Guided KV Cache Compression"
_ICML.cc/2026/Conference — ICML 2026 regular_

### Official Review · Reviewer_eVYW · 2026-02-16

**Soundness:** 3
**Presentation:** 3
**Significance:** 3
**Originality:** 3
**Overall Recommendation:** 4
**Confidence:** 4

**Summary:**

RLKV is a KV cache compression method for long-cot LLMs by using RLVR to measure how dynamic compression affects reasoning performance. With an L1 sparsity penalty, it identifies a subset of reasoning-critical attention heads that retain a full KV cache while aggressively compressing the rest. This targeted strategy reduces KV memory by 20–50% with near-lossless accuracy and some minor wall-clock improvements.

**Compliance With Llm Reviewing Policy:**

Affirmed.

**Final Justification:**

The main ideas of this paper is valuable.

However, it remains that experiments (training part) were only conducted on a single domain, which weakens evidence for this method as a general one.

Also, please provide average values for the shared tables so that performance comparison is clear.

Authors are strongly encouraged to add these revisions into the revised PDF.

Accordingly, the existing score seems appropriate. Confidence 3 --> 4.

**Key Questions For Authors:**

See weaknesses

**Strengths And Weaknesses:**

Strengths

(1) The problem setting is important, and the reported efficiency gain is solid.

(2) A one-off training cost to improve inference efficiency is the right direction for large-scale use cases.

(3) The presentation is generally strong, while some of the small text in figures and tables is hard to read (minor modifications can be made). Please note that the entire main body should be legible when printed on A4 paper.

(4) While the related work discussion is solid, it is unclear if all relevant and recent methods have been included as baselines (e.g., PruLong). This is further discussed in the weaknesses.

Weaknesses

(1) There is only one baseline, DuoAttention (2024), for “Head-reallocation,” which makes it the most relevant for the paper. Are the authors sure that there are no other recent baselines? This is especially critical as the authors claim “state-of-the-art.”

(2) A discussion on the following would help: Does this method work for various hardware configurations? For instance, a multi-GPU (or multi-node) setup? How would it dynamically work for GPUs with different VRAM capacities? These can change depending on the inference setting and as hardware evolves.

(3) Is this method fully compatible with GQA? Do all the baselines embody common speed-ups like GQA? Could the authors provide a discussion regarding its compatibility?

(4) The method introduces numerous new hyperparameters, yet there are no sensitivity tests. Additionally, it may be cumbersome to tune these hyperparameters (e.g., “Regularization weight,” “Reward threshold,” “Sink token size,” and “Local token size”).

(5) Line 165: "In concrete" should be changed to "Concretely,".

(6) It is unfortunate that training is only done on the math domain.

(7) In Fig. 1 (b) and Fig. 9, the more transparent “repetitive” and “incorrect” parts are very hard to discern. I suggest the authors reconsider using transparency to distinguish parts of the stacked bar charts.

(8) Authors are encouraged to discuss key limitations and weaknesses. If space is a limiting factor, it should at least be in the Appendix.

---

> ### Author Rebuttal · Authors · 2026-03-29
>
> We sincerely thank the reviewer for the thoughtful feedback and constructive suggestions. Below, we address each point.
>
> ---
> - W1: Baseline Comparisons
>
> We acknowledge the concern and have expanded our experiments to include KVZip[1] (NeurIPS 2025), which provides head-reallocation variants and claims to surpass DuoAttention in long-context tasks.
> KVZip uses maximum token-level attention scores as a proxy for head importance when prompting the model to repeat the context in a single prefill.
>
> Table 1: on Llama-3.1-8B-R1; 0.2-0.8 denotes sparsity levels.
> |Benchmark|Method|Full|0.2|0.4|0.6|0.8|
> |-|-|-|-|-|-|-|
> |GSM8K|DuoAttn|89.16|88.78|87.11|77.79|28.51|
> ||KVZip||89.23|81.8|78.32|34.27|
> ||RLKV||89.23|86.81|79.45|47.38|
> |Math500|DuoAttn|83.0|84.4|81.6|69.0|28.6|
> ||KVZip||79.0|75.4|65.4|24.4|
> ||RLKV||84.4|84.6|78.0|49.6|
> |AIME24|DuoAttn|36.67|33.33|20.0|6.67|0.0|
> ||KVZip||30.0|20.0|13.33|3.33|
> ||RLKV||36.67|40.0|20.0|0.0|
> |MBPP|DuoAttn|62.6|62.0|60.6|40.4|7.4|
> ||KVZip||59.8|57.8|39.6|12.6|
> ||RLKV||62.8|63.8|51.8|6.0|
> |Chem.|DuoAttn|45.8|43.2|36.4|30.4|3.8|
> ||KVZip||40.6|38.6|0.2|0.0|
> ||RLKV||44.4|44.8|37.8|2.2|
> |CS|DuoAttn|51.46|48.29|46.1|34.39|9.76|
> ||KVZip||47.8|40.49|2.93|0.24|
> ||RLKV||49.27|47.56|43.17|5.12|
> |Law|DuoAttn|26.4|24.2|19.8|13.0|3.6|
> ||KVZip||16.6|15.6|0.8|0.8|
> ||RLKV||23.6|22.8|20.6|4.2|
> |Phys.|DuoAttn|44.0|40.6|38.6|29.4|6.6|
> ||KVZip||38.4|36.2|1.2|0.0|
> ||RLKV||40.8|42.8|37.4|4.4|
>
> Table 2: on Math500
> |Model|Method|Full|0.2|0.4|0.6|0.8|
> |-|-|-|-|-|-|-|
> |Llama-3.1-8B-R1|DuoAttn|83.0|84.4|81.6|69.0|28.6|
> ||KVZip||79.0|75.4|65.4|24.4|
> ||RLKV||84.4|84.6|78.0|49.6|
> |Qwen-2.5-7B-R1|DuoAttn|87.8|83.6|74.2|56.4|8.4|
> ||KVZip||87.0|81.6|60.8|3.4|
> ||RLKV||89.0|86.0|71.4|15.8|
> |Qwen-3-4B-Thinking|DuoAttn|77.6|78.2|75.4|53.8|23.6|
> ||KVZip||79.2|83.0|71.2|1.0|
> ||RLKV||77.8|83.2|75.6|13.2|
>
> Notably, KVZip underperforms DuoAttention in reasoning models, despite being a more recent method.
> This suggests that a single-pass, reasoning-agnostic proxy is insufficient to identify reasoning-critical heads, supporting RLKV's design of training over diverse data with on-policy RL reward signal.
> These results further confirm RLKV's effectiveness.
>
> We additionally evaluate KVZip on the long-context reasoning benchmark LongReason (Table 1 in our response to Reviewer `8QBD`).
> We also provide the SFT ablation (Table 1&2 in our response to Reviewer `BcwG`), which trains gating adapters with CE loss, serves as a simplified variant of PruLong.
> RLKV consistently outperforms both extra experiments.
>
> [1] KVzip: Query-Agnostic KV Cache Compression with Context Reconstruction, NeurIPS 2025
>
> ---
> - W2: Hardware Compatibility
>
> RLKV produces a static gating adapter that is fully independent of hardware configuration.
> As discussed in Section 4.3, deployment requires customized attention and KV cache management in software.
> For different VRAM budgets, no dynamic adaptation is needed: the sparsity ratio is a deployment-time decision that directly controls KV cache usage. For multi-GPU and multi-node setups, tensor parallelism in attention naturally partitions heads across GPUs, and RLKV's gating adapter operates at the head level.
>
> ---
> - W3: GQA Compatibility
>
> Llama-3.1-8B-R1, Qwen-2.5-7B-R1, and Qwen-3-4B-Thinking are all GQA architectures. RLKV operates at the KV head level and is naturally compatible with GQA — no special handling is needed.
>
> ---
> - W4: Hyperparameter Sensitivity
>
> RLKV is designed to be robust and easy to deploy, introducing only two new tunable hyperparameters: regularization weight and reward threshold, both ablated in Section 4.4.
> Other training settings are fixed.
> We will clarify this distinction between tunable and fixed parameters in the revised manuscript.
>
> ---
> - W6: Training Domain
>
> We chose math as the training domain because it provides the cleanest verifiable reward signal for RL, and RLKV uses reward signals to capture reasoning behavior rather than overfitting to the training domain.
> As shown in our original experiments, RLKV demonstrates strong generalization on code generation (MBPP) and knowledge QA (MMLU-Pro).
> Furthermore, RLKV is trained within 8K context yet generalizes to 70K long-context reasoning (Table 1 in our response to Reviewer `8QBD`), further confirming that it captures general reasoning behavior rather than domain-specific patterns.
>
> ---
> - W8: Limitations
>
> We appreciate the suggestion. We briefly outline the key limitations below and will provide a thorough discussion in the revised manuscript:
>  1. Further inference speedup is possible with customized attention kernels for serving frameworks (Section 4.3).
>  2. The computational cost of RL rollouts during training remains a challenge; off-policy RL strategies could help reduce this cost.
>
> ---
> - W5 & W7
>
> We appreciate the reviewer pointing out inaccuracies. We will revise the manuscript to improve clarity and readability.
>
> ---
> We hope our response addresses all concerns. We are actively available during the next Author-Reviewer Discussion.

---

> > ### Author Rebuttal · Reviewer_eVYW · 2026-04-02
> >
> > Thanks to the authors for their effort.
> >
> > The main ideas of this paper is valuable.
> >
> > However, it remains that experiments (training part) were only conducted on a single domain, which weakens evidence for this method as a general one.
> >
> > Also, please provide average values for the shared tables so that performance comparison is clear.
> >
> > Authors are strongly encouraged to add these revisions into the revised PDF.
> >
> > Accordingly, the existing score seems appropriate. Confidence 3 --> 4.

---

> > > ### Author Response · Authors · 2026-04-04
> > >
> > > We thank the reviewer for recognizing the value of our main ideas and for the constructive feedback throughout. We appreciate the reviewer's time.
> > >
> > > We provide the requested averages below.
> > >
> > > Average across benchmarks on Llama-3.1-8B-R1 (from Table 1):
> > > |Method|Full|0.2|0.4|0.6|0.8|
> > > |-|-|-|-|-|-|
> > > |DuoAttn|57.62|52.85|47.72|29.65|3.54|
> > > |KVZip||54.96|47.97|29.92|1.10|
> > > |RLKV||59.16|53.92|41.81|3.73|
> > >
> > > Average across models on Math500 (from Table 2):
> > > |Method|Full|0.2|0.4|0.6|0.8|
> > > |-|-|-|-|-|-|
> > > |DuoAttn|82.80|82.07|77.07|59.73|20.20|
> > > |KVZip||81.73|80.00|65.80|9.60|
> > > |RLKV||83.73|84.60|75.00|26.20|
> > >
> > > RLKV consistently outperforms both baselines across all sparsity levels in both averaged settings.
> > >
> > > As for the training domain concern, we respectfully note that generalization is best measured by cross-domain evaluation, not by training-domain diversity.
> > > RLKV trained on math alone generalizes well to code (MBPP), law, physics, chemistry, CS, and long-context multi-hop reasoning (LongReason), with no sign of domain-specific overfitting.
> > > This is consistent with established practice — DuoAttention[1], RazorAttention[2], and KVZip[3] all calibrate on a single domain and validate generalization through cross-domain evaluation.
> > > That said, exploring how training domain composition affects adapter learning is an interesting direction for future work.
> > >
> > > All suggested revisions will be incorporated into the revised PDF.
> > >
> > > [1] DuoAttention: Efficient Long-Context LLM Inference with Retrieval and Streaming Heads, ICLR 2025.
> > > [2] RazorAttention: Efficient KV Cache Compression Through Retrieval Heads, ICLR 2025.
> > > [3] KVZip: Efficient Key-Value Compression for Long Contexts, NeurIPS 2025.

---

### Official Review · Reviewer_BcwG · 2026-03-12

**Soundness:** 3
**Presentation:** 4
**Significance:** 2
**Originality:** 2
**Overall Recommendation:** 4
**Confidence:** 2

**Summary:**

The paper proposes to sparsify the attention by removing redundant KV heads and preserving valuable heads for tasks involved in reasoning and inference. The paper does that by adding a special trainable gating adapter, which is trained by both the objective loss and an L1 norm to increase sparsity. Also, the paper uses GRPO to train the model to focus on reasoning tasks. The paper shows that the trained sparse model is better in performance than prior static sparsity methods under various reasoning tasks. Also, the paper shows an efficiency gain over dense attention.

**Compliance With Llm Reviewing Policy:**

Affirmed.

**Final Justification:**

Hi, I think the paper presents a concrete method for training a sparse attention gating function. The method is thoroughly studied and ablated, and the paper shows practical speedup. However, I think that although the authors attempt to justify the necessity of the RL and SFT in the rebuttal, it still lacks more analysis on the area where the rebuttal's tight text space might limit. For that, I decided to keep the score. I encourage the authors to present more intuition and analysis in the paper going further.

**Key Questions For Authors:**

1. What would happen if you only do SFT on reasoning traces for the training parts? What does RL bring to identifying the sparsity pattern that SFT cannot find?
2. How fast is the convergence? How many GPU hours are required for training? Do you need special data mixture or just math data?
3. What is the motivation behind KV heads that are suitable for reasoning are static? Any evidence that KV heads might behave different across different prompt, maybe sometimes important but sometimes not too much?

**Limitations:**

Yes

**Strengths And Weaknesses:**

Strengths
1. The paper identifies the presence of KV heads that are more suitable for reasoning tasks and proposes a way to identify them
2. The paper presents concrete results that beat prior related methods and are consistent across different benchmarks
3. The paper presents comprehensive evaluations with various benchmarks for performance and presents end-to-end latency for efficiency gain.

Weakness
1. The paper doesn't clearly showcase the necessity of RL in the method. Can SFT on reasoning traces also expose the reasoning-favorable KV heads? If so, SFT is faster and cheaper.
2. The static sparsity methods presented by the paper and prior works don't require training, while the paper does. There is limited discussion on how long the training requires and what happens if not trained sufficiently or excessively.
3. The speedup presented by the paper isn't substantial.

---

> ### Author Rebuttal · Authors · 2026-03-29
>
> We sincerely thank the reviewer for the thoughtful feedback and constructive suggestions. Below, we address each point.
>
> ---
> - W1 & Q1: SFT ablation study
>
> We appreciate the reviewer's suggestion. To address this, we first explain why RL is better suited than SFT for learning gating adapters, then validate this with an ablation study.
>
> Gating adapters alter the attention output and effectively change the model's policy. RL is naturally suited here: its rollouts are generated under this adapted policy, so the reward signal directly reflects reasoning quality with sparsified adapters.
> In contrast, SFT trains the adapted model to replicate reasoning traces produced by the original, unadapted model. The reasoning traces were produced under full attention and may not represent effective strategies under the sparse regime.
> Forcing the adapted model to imitate them therefore hurts both performance and generalization.
>
> To verify this, we conduct an SFT ablation that replaces the RL objective with CE loss while keeping all other settings identical, including the L1 penalty, the same 3K reasoning traces, and training epochs.
> Since SFT converges much faster and pushes adapter values lower than RLKV does, we use a lower learning rate and a smaller regularization weight for best performance.
> Training takes ~2 hours on a single A100 80G, faster than RLKV on Llama-3.1-8B-R1.
>
> Table 1: on Math500 (in-domain); 0.2-0.8 denotes sparsity levels.
> |Method|Full|0.2|0.4|0.6|0.8|
> |-|-|-|-|-|-|
> |DuoAttention|83.0|84.4|81.6|69.0|28.6|
> |RLKV||84.4|84.6|78.0|49.6|
> |SFT||77.0|60.0|27.8|1.8|
>
> Table 2: on MMLU-Pro-Chem (out-of-domain)
> |Method|Full|0.2|0.4|0.6|0.8|
> |-|-|-|-|-|-|
> |DuoAttention|51.6|47.4|44.8|24.8|2.2|
> |RLKV||55.0|50.8|36.0|0.4|
> |SFT||37.8|19.0|4.0|0.4|
>
> SFT significantly underperforms RLKV across all sparsity levels on both benchmarks.
> On Math500, SFT collapses at 0.6 sparsity (27.8% vs. 78.0%), where fewer heads retain full KV cache and correct allocation becomes critical.
> This suggests that SFT's head importance ranking reflects the dense model's preferences rather than the true needs under the sparse regime, leading to misallocation.
> The gap is even more pronounced out-of-domain: on MMLU-Pro-Chem, SFT already collapses at 0.2 sparsity (37.8% vs. 55.0%), indicating that the distribution mismatch not only degrades performance but severely limits generalization.
>
> ---
>
> - W2 & Q2: Training Setting
>
> We provide details in Appendix B. On 2×A100 GPUs, RLKV takes 11–20 hours depending on the model. This is a one-off cost that benefits all subsequent inference.
>
> Insufficient training leaves adapter values under-separated.
> The top-k selection then cannot cleanly distinguish critical heads from compressible ones, degrading performance at high sparsity.
> Excessive training is harmless. Our adaptive penalty stops penalizing a head once its adapter falls below the reward threshold.
> Reward then fluctuates mildly near the threshold and the adapter distribution stabilizes.
>
> With only 3K math problems (collected via self-distillation, no special data mixture), training converges stably within 2 epochs.
> Our data recipe further supports our generalization claims.
>
> ---
> - W3: Speedup
>
> We use a vanilla implementation to functionally verify RLKV's effectiveness: reduced KV cache usage and wall-clock speedup.
>
> End-to-end decoding acceleration is a complex systems engineering problem.
> Frameworks like SGLang and FlashInfer represent substantial engineering efforts with customized attention kernels dedicated to this goal.
> Head-reallocation methods (including ours) manage two types of KV cache (full and compressed) per head, which could be compatible with these frameworks but requires customized kernels to achieve ideal acceleration.
>
> ---
> - Q3: Static Head Importance Scores
>
> RLKV's adapter values are trained on 3K diverse CoT traces. The scores reflect a global ranking of head importance, not a single-prompt signal.
>
> We use static scores at inference for efficiency. Query-aware allocation would require dynamic per-query decisions, breaking head-level parallelism (exactly the issue faced by HeadKV [1]).
>
> Empirically, our cross-domain results confirm static scores is effective: math-trained adapters generalize to code (MBPP), knowledge QA (MMLU-Pro), and long-context reasoning (Table 1 in our response to Reviewer `8QBD`).
>
> Prior work on retrieval heads [2] shows the strongest retrieval heads are consistently activated across contexts, while weaker ones are query-aware.
> However, that analysis targets factual retrieval in standard MHA LLMs.
> Whether the same holds for reasoning-critical heads in reasoning models is an open question.
>
> [1] Not All Heads Matter: A Head-Level KV Cache Compression Method with Integrated Retrieval and Reasoning, ICLR 2025.
> [2] Retrieval Head Mechanistically Explains Long-Context Factuality, ICLR 2025.
>
> ---
> We hope our responses address all concerns. We are actively available during the next Author-Reviewer Discussion period.

---

> > ### Author Rebuttal · Reviewer_BcwG · 2026-04-04
> >
> > I hence maintain my positive score.

---

> > > ### Author Response · Authors · 2026-04-04
> > >
> > > We sincerely thank the reviewer for acknowledging the resolution and for the constructive feedback throughout. We appreciate the reviewer's time.
> > >
> > > As a follow-up to W3, we implement RLKV in SGLang with a dual KV pool architecture, where full-attention heads use the standard paged KV cache and compressed heads use a fixed-size circular buffer (sink + local window). Freed memory is rebalanced to increase request concurrency.
> > >
> > > Results on MATH-500 (Llama-3.1-8B-R1, single A100 40G, SGLang v0.5.2):
> > >
> > > | Sparsity | Accuracy | Theoretical | Time (s) | E2E Speedup | Throughput (tok/s) | Thpt Speedup |
> > > |----------|----------|-------------|----------|-------------|--------------------|--------------|
> > > | 0% (Full)| 79.4     | —           | 1036     | 1.00x       | 1500               | 1.00x        |
> > > | 20%      | 78.6     | 1.24x       | 874      | 1.19x       | 1758               | 1.17x        |
> > > | 40%      | 79.6     | 1.64x       | 666      | 1.56x       | 2185               | 1.46x        |
> > > | 50%      | 77.6     | 1.95x       | 574      | 1.80x       | 2553               | 1.70x        |
> > > | 60%      | 73.8     | 2.40x       | 504      | 2.06x       | 2941               | 1.96x        |
> > >
> > > Theoretical speedup = 1/((1−s) + s·W/L), where W is the compressed window size (sink + local = 80 tokens) and L is the average sequence length (~3000 tokens under continuous batching). This accounts for the residual memory cost of compressed heads. This implementation achieves near-theoretical speedup at practical sparsity levels. The remaining gap to theoretical is due to the dual-dispatch attention overhead; customized fused kernels for heterogeneous head attention can further close this gap.
> > >
> > > This engineering effort further validates the practical value of RLKV's head-reallocation strategy for real-world serving.

---

### Official Review · Reviewer_8QBD · 2026-03-16

**Soundness:** 3
**Presentation:** 3
**Significance:** 3
**Originality:** 3
**Overall Recommendation:** 5
**Confidence:** 4

**Summary:**

The paper present an RL-guided KV-cache compression methodology called RLKV which identifies which heads contribute to reasoning and keeps those heads in full while compressing the rest of the heads. Authors provide detailed evaluations and show the speed benefits of their approach.

**Compliance With Llm Reviewing Policy:**

Affirmed.

**Key Questions For Authors:**

NA

**Limitations:**

Yes

**Strengths And Weaknesses:**

Strengths:
 - The paper is well written and easy to follow.
 - The methodology is simple and well motivated. Reasonable evaluations and assessment is provided.
 - Their approach is novel and addresses an important problem in the field.

Weaknesses:
 - Although authors covered some tasks in their evaluations, it is not clear how their approach generalizes to other tasks such as agentic ones as well as tasks that require long context.

---

> ### Author Rebuttal · Authors · 2026-03-29
>
> We sincerely thank the reviewer for the positive assessment. Below, we address the generalization concern.
>
> ---
> - W1: Generalization to Other Tasks
>
> **Long-context generalization.** We further evaluate on LongReason[1], a challenging long-context reasoning benchmark that requires models to perform reading comprehension, logical inference, and mathematical word problems over long inputs. We use its 64K-input subset (400 randomly sampled from 794 instances), set the model context length to 70K, and evaluate Llama-3.1-8B-R1 across four sparsity levels for R-KV, DuoAttention, KVZip[2], and RLKV. H2O runs out of memory in this setting, and KVZip is the recent head-reallocation baseline suggested by Reviewer `eVYW`.
>
> Table 1: on LongReason-64K (70K context); 0.2-0.8 denotes sparsity levels.
> |Method|Full|0.2|0.4|0.6|0.8|
> |-|-|-|-|-|-|
> |R-KV|49.25|0.0|0.0|0.0|0.0|
> |DuoAttention||49.5|48.75|35.25|1.5|
> |KVZip||48.0|49.0|36.0|4.75|
> |RLKV||50.5|52.5|45.25|15.0|
>
> Three observations:
> 1. RLKV substantially outperforms all baselines on long-context reasoning, especially at high sparsity levels.
> 2. Head-reallocation methods (DuoAttention, KVZip) can preserve reasoning ability, since they retain full KV cache for a subset of heads, providing support for reasoning behavior. However, as sparsity increases, their performance degrades more steeply than RLKV, because their head identification relies on proxy objectives that do not directly capture reasoning behavior.
> 3. The token-dropping method R-KV collapses entirely at all sparsity levels, since every output degenerates into a repetitive loop. This again confirms that token-dropping inevitably disrupts CoT consistency.
>
> Notably, the training data used in RLKV's self-distillation sampling consists of questions which can be solved under 8K tokens. The fact that RLKV generalizes to 70K contexts demonstrates that it captures reasoning behavior itself, which transfers long-context scenarios.
>
> **Agentic tasks.** Extending RLKV to agentic tasks is a promising direction, but two gaps make evaluation premature at this stage.
>
> First, small reasoning LLMs exhibit limited agentic capability. Recent agentic benchmarks [3-4] show that only the largest frontier models perform well on these tasks.
> Evaluating KV cache compression on models that themselves struggle with these tasks would offer limited insight.
>
> Second, agentic reasoning differs fundamentally from closed-world reasoning [5]. It plans, acts, and learns through continual interaction with open-ended and dynamic environments, which goes beyond the reasoning behaviors of traditional models.
> As a result, applying RLKV to agentic tasks would require retraining adapters on agentic-capable models with agentic task data. The research community does not yet support this.
>
> We believe RLKV can be applied to agentic reasoning once these prerequisites are met. Agentic behaviors also require revisiting past KV cache (e.g., recalling tool outputs from earlier turns), so we hypothesize that certain heads will emerge as agentic-critical. We view this as an important future direction.
>
> [1] LongReason: A Synthetic Long-Context Reasoning Benchmark via Context Expansion, arXiv 2025.
> [2] KVzip: Query-Agnostic KV Cache Compression with Context Reconstruction, NeurIPS 2025.
> [3] BrowseComp: A Simple Yet Challenging Benchmark for Browsing Agents, https://llm-stats.com/benchmarks/browsecomp
> [4] WideSearch: Benchmarking Agentic Broad Info-Seeking, https://widesearch-seed.github.io/#leaderboard
> [5] Agentic Reasoning for Large Language Models, arXiv 2026.
>
> ---
> We hope this response addresses your concern. Please do not hesitate to share further questions or suggestions. We are actively available during the next Author-Reviewer Discussion period.

---

> > ### Author Rebuttal · Reviewer_8QBD · 2026-04-01
> >
> > No major concern

---

> > > ### Author Response · Authors · 2026-04-04
> > >
> > > We thank the reviewer for the positive assessment and for confirming that the concerns have been addressed. We appreciate the reviewer's time.

---

### Decision · Program_Chairs · 2026-04-30

**Decision:**

Accept (regular)

**Comment:**

This paper proposes RLKV, a method that uses reinforcement learning to identify which attention heads are critical for reasoning in LLMs, enabling effective KV cache compression by allocating full cache to important heads while aggressively compressing others. The reviewers unanimously recognized the importance of the problem and the novelty of the approach, with scores of 5 (Accept), 4 (Weak Accept), and 4 (Weak Accept). Key concerns included the necessity of RL over SFT, generalization beyond the math training domain, limited baselines, and modest speedups. The authors provided thorough rebuttals addressing each point: an SFT ablation convincingly demonstrated RL's superiority, additional baselines confirmed state-of-the-art performance, long-context experiments showed strong generalization from short-context training, and an optimized serving implementation demonstrated practical speedups. Two of three reviewers acknowledged their concerns as fully resolved, and the third maintained a positive score while requesting minor revisions. The paper is well-written, tackles a timely problem at the intersection of reasoning LLMs and inference efficiency, and offers a principled and empirically validated solution. I recommend acceptance.